# Lower plate serpentinite diapirism in the Calabrian Arc subduction complex

A. Polonia [1], L. Torelli[2], L. Gasperini[1], L. Cocchi [3], F. Muccini[3], E. Bonatti[1,4], C. Hensen[5], M. Schmidt[5], S. Romano [1], A. Artoni [2] & M. Carlini [2]

Mantle-derived serpentinites have been detected at magma-poor rifted margins and above subduction zones, where they are usually produced by fluids released from the slab to the mantle wedge. Here we show evidence of a new class of serpentinite diapirs within the external subduction system of the Calabrian Arc, derived directly from the lower plate. Mantle serpentinites rise through lithospheric faults caused by incipient rifting and the collapse of the accretionary wedge. Mantle-derived diapirism is not linked directly to subduction processes. The serpentinites, formed probably during Mesozoic Tethyan rifting, were carried below the subduction system by plate convergence; lithospheric faults driving margin segmentation act as windows through which inherited serpentinites rise to the sub-seafloor. The discovery of deep-seated seismogenic features coupled with inherited lower plate serpentinite diapirs, provides constraints on mechanisms exposing altered products of mantle peridotite at the seafloor long time after their formation.

[1] Institute of Marine Sciences CNR ISMAR-Bo, Via Gobetti, 101, 40129 Bologna, Italy. [2] Department of Chemistry, Life Sciences and Environmental Sustainability, University of Parma, Parco Area delle Scienze, 157/A Parma, Italy. [3] Istituto Nazionale di Geofisica e Vulcanologia, via di Vigna Murata 605, 00143 Rome, Italy. [4] Lamont-Doherty Earth Observatory of Columbia University, Palisades, NY 10964, USA. [5] GEOMAR Helmholtz-Zentrum für Ozeanforschung Kiel, Wischhofstrasse 1-3, 24148 Kiel, Germany. Correspondence and requests for materials should be addressed to A.P. (email: alina.polonia@ismar.cnr.it)

Tectonic activity in subduction-rollback convergent plate boundaries commonly involves backward migration of subducting slabs, arc magmatism, and plate boundary segmentation. Incipient collision, the presence of indentors, and generally lower plate topography may increase complexity and plate fragmentation. Slab tearing has a major role in developing segmented subduction zones, in particular in narrow slab segments[1], triggering asthenospheric upwelling, dynamic topography[2], and magmatism[3,4].

The origin of the Ionian Sea is related to the opening of the Neo-Tethys basin[5], which has now mostly disappeared below the Alps/Apennines. An exceptionally thick sedimentary cover (up to 6–8 km) and the lack of detailed magnetic anomalies makes it difficult to define the nature of the Ionian Sea crust, which has been interpreted as either oceanic[6–8] or oceanic/severely stretched continental[9,10], or thinned continental[11–13].

Consumption of the Neo-Tethys lithosphere since Oligocene times has caused the emplacement of the Calabrian Arc (CA), a narrow and arcuate subduction system (Fig. 1) related to Africa/Eurasia plate convergence, and to the southeastward retreat of the Tethyan slab[14]. Convergence generated a 10–30 km-thick, 300 km-wide subduction system, including a submarine accretionary wedge[15]. The migrating Calabria trench drives the entire subduction complex outward, but different boundary conditions produce segmentation in two distinct lobes, with different structural and morphotectonic characteristics (Fig. 1). The eastern lobe (EL) collides with the Hellenic subduction system and produces basement-involved tectonics similar to a fold and thrust belt[15], whereas the western lobe (WL) is free to spread into the abyssal plain of the Ionian Sea (Fig. 1) through frontal accretion and offscraping processes[16].

Two oppositely dipping fault systems, i.e., the Ionian fault (IF) and Alfeo-Etna fault (AEF), deform the WL offshore Eastern Sicily (Fig. 1) along a complex strike-slip/transtensional pattern[15,17]. These two systems mark the external boundary of a wide deformation zone including three additional faults (F1, F2, and F3; Fig. 1) and they might be the source of major earthquakes in the CA[18]. Trans-tensional deformation and deep fragmentation of the Western Ionian domain[19–21] are in agreement with geodetic measurements[22], suggesting that Calabria is moving SE and Southern Sicily is moving South in an Apulia-fixed reference frame. Moreover, mechanical/numerical models taking into account geometry and strength contrast across the Neo-Tethyan passive margin and major faults, suggest that pre-existing fracture zones and transform faults controlled the subsequent Ionian margin segmentation[17]. In this context, the NW portion of the IF corresponds to basement deformation associated with slab tearing, whereas the AEF might be the result of regional scale dextral shearing.

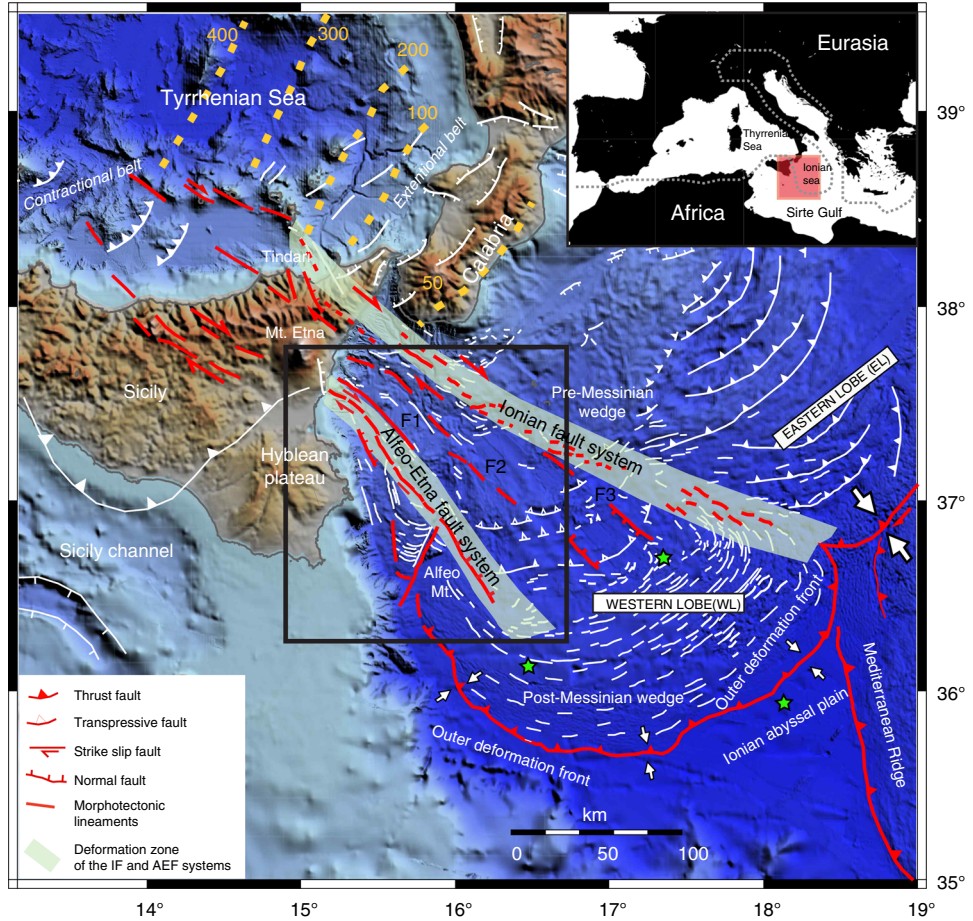

**Fig. 1** Geodynamic setting of the study area represented by the black box and the red area in the inset map. The structural model is modified from ref. [17]. Red lines: major faults absorbing plate motion; white lines: second order faults. The north-western (NW) ward dipping subducting slab of the African plate is represented by yellow iso-depth lines in the Tyrrhenian Sea. Major structural boundaries, active faults and extent of structural domains are indicated. The continental margin is segmented along the Ionian fault (IF) and Alfeo-Etna fault (AEF) systems whose deformation zone is represented by a light blue pattern. F1, F2, F3: transtensional faults belonging to the wide corridor of deformation between the AEF and IF[17]. White arrows represent shortening along the outer deformation front (higher rates in the EL) of the subduction systems. Green stars: location of OBSs used for velocity models of the lithosphere[26]

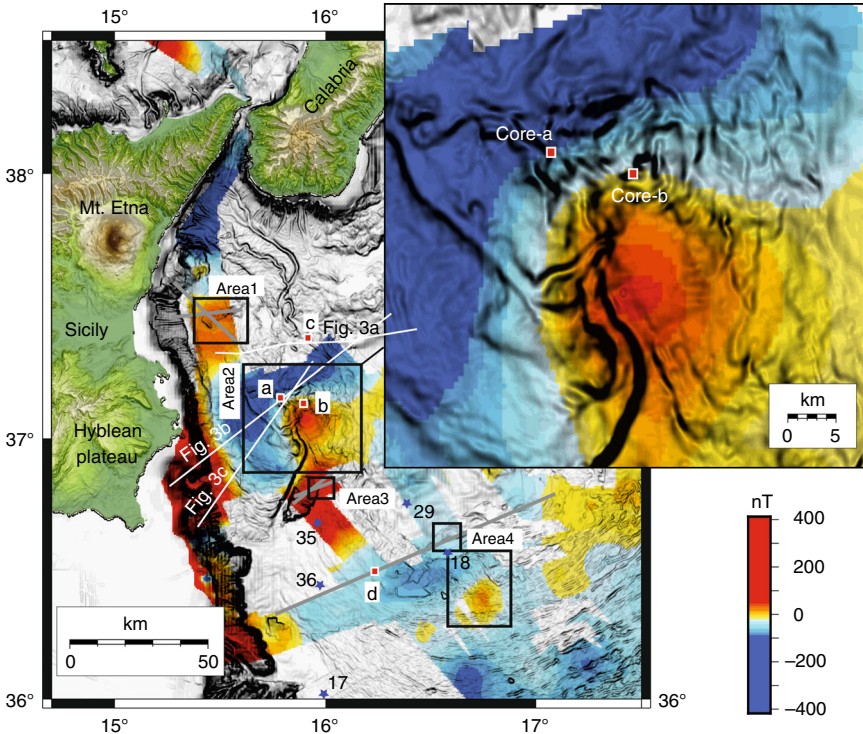

**Fig. 2** Magnetic anomaly map derived from this study superimposed over a gray levels bathymetric slope map. The four major areas of mantle diapirism described in this work are indicated (black squares 1–4). The inset highlights the rough seafloor topography of Area-2, which corresponds to the magnetic anomaly. White lines correspond to MCS data shown in Fig. 3, whereas gray lines are shown in the supplementary material. Blue stars represent heat flow measurements from ref. [23] (17: $15.7 \pm 1.9$ mW m$^{-2}$; 18: $16.2 \pm 2.7$ mW m$^{-2}$; 29: $26.1 \pm 5.2$ mW m$^{-2}$; 35: $30.3 \pm 5.0$ mW m$^{-2}$; 36: $34.4 \pm 4.0$ mW m$^{-2}$). Red squares **a**–**d** represent gravity cores described in the manuscript (Fig. 7) and Supplementary Figs. 9–12

Seismic reflection profiles integrated with multibeam and potential field data reveal 13 sub-circular magnetic bodies beneath the seafloor aligned along transtensional faults segmenting the continental margin. We investigated the nature of such bodies within the accretionary wedge, their link with Mesozoic paleoceanography and with the latest Quaternary extensional phase that disrupted the subduction complex. Our multidisciplinary approach suggests that the diapirs are made of inherited Mesozoic Tethyan serpentinites, which rise along transtensive faults controlling rifting processes perpendicular to the subduction arc. They provide the first example of serpentinite diapirism from the lower plate in a subduction system. Mantle-derived serpentinites combined with geophysical data suggest that the ocean-continent transition (OCT) in the Ionian Sea has a complex geometry reflecting the 90° curvature of the continental margin. Oceanic crust, if present, should be limited to the inner accretionary wedge of the subduction complex and the AEF should be located close to the Mesozoic Tethyan OCT and/or to a major transform fault.

## Results

**Rifting and diapirism along the AEF system.** Multibeam, seismic reflection, magnetic, and gravity field data call for incipient deformation of four areas marked by buried sub-circular features aligned along the AEF (Fig. 2). Area-1 is offshore Mt. Etna, whereas Area-2 and -3 are located north and northeast of Alfeo Seamount; the more distal Area-4 is located on a slope terrace of the accretionary complex. Deep deformation in these areas is imaged by seismic reflection lines showing sediment tilting and up warping around acoustically blind nuclei (Fig. 3). More information can be found in Supplementary Figs. 1–7. Multibeam

data show sub-circular seafloor patterns, marked by increased roughness particularly in Area-2 (Fig. 2).

Deep penetrating multichannel seismic lines collected in Areas-1 and -2 (Fig. 3) run across the subsiding WL of the subduction complex, from the Malta escarpment to the IF, and image crustal trans-tensional faults segmenting the accretionary wedge. The AEF forms at the boundary between the salt-bearing complex emplaced after the Messinian salinity crisis and the older wedge consisting of clastic sediments (Fig. 3b). This fault system served as main thrust fault during Messinian time, when frontal accretion resumed after the desiccation of the Mediterranean Sea[15], and later accommodated transtensive deformation, as indicated by a syn-tectonic sedimentary basin in the footwall (Fig. 3b). Corresponding to the AEF, the basal detachment is disrupted, displaced and associated with large 10–20 km-wide diapirs. A reflector at about 13 s two-way travel time (TWT) in seismic line ETNASEIS 6 and 11 s (TWT) in lines CA-A is interpreted as the Moho (Fig. 3a, b). Arching and disruption of deep reflectors in correspondence to the diapiric structures suggest that the source region is deep and below the detachment.

Rising material in subduction complexes might be ascribed to different processes, such as salt/mud diapirism, magmatic intrusions, and serpentinite diapirism. We attempted to discriminate between these processes by gravity and magnetic modeling (Supplementary Fig. 8). A salt source for the diapirs can be excluded because no thick salt was deposited in the inner wedge during Messinian times[15], and because the source of the rising material is deeper than the Messinian salt layer (Fig. 3). Moreover, salt and mud diapirs would not cause positive magnetic anomalies; gravity modeling also excludes a mud composition for the diapirs (Supplementary Fig. 8). As these structures are aligned offshore Mt. Etna, an origin related to

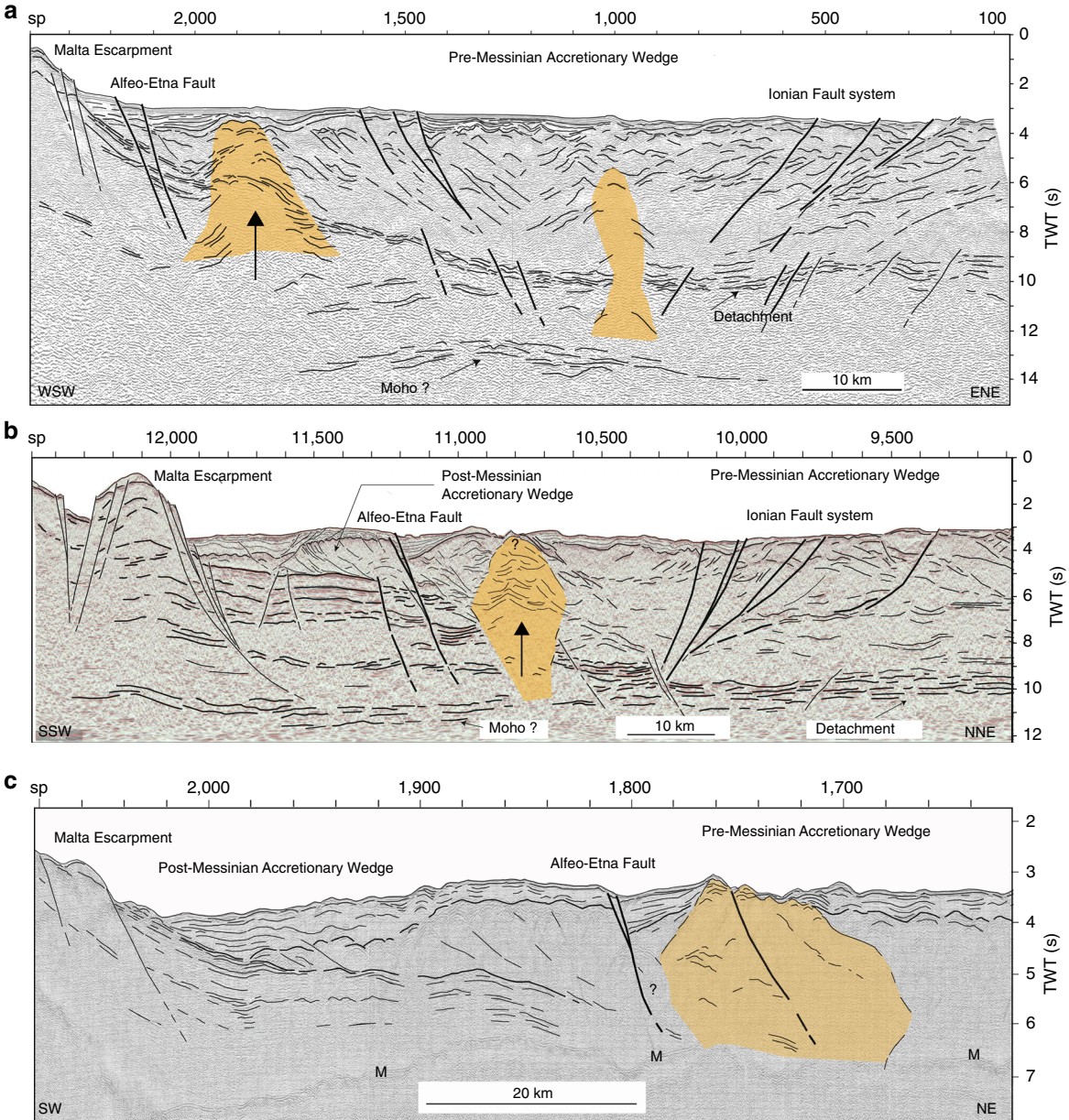

**Fig. 3** Line drawings of time migrated MCS lines used for structural reconstructions in Areas-1 and -2. Location of seismic profiles is shown in Fig. 2. Yellow/brown pattern: diapiric features. **a** ETNASEIS-6 MCS line collected orthogonal to the continental margin SE of Etna. This line crosses the entire WL of the accretionary wedge intercepting both the Alfeo-Etna and Ionian faults. Such faults control extensional processes, which are associated with a shallower Moho at about 11 s TWT. A second diapiric feature is present in the centre of the rifting zone. **b** CA-A MCS line collected across the WL of the Calabrian Arc subduction complex orthogonally to the AEF. A reflector at about 11 s TWT is interpreted as the Moho reflection. **c** MS-26 MCS line across the AEF north of the Alfeo Seamount. M, multiple. The diapir corresponds to a very rough seafloor morphology. Bold lines: major trans-tensional faults controlling rifting processes

magmatism needs to be explored. However, magnetic and gravity anomalies are not consistent with a volcanic or magmatic source (Supplementary Fig. 8) in agreement with heat flow data showing a lower than normal thermal regime for the Ionian basin (mean heat flow value is $31.8 \pm 5.0$ mW m$^{-2}$ [23]) (Fig. 2). As the basal detachment of the subduction system is deformed and segmented in correspondence with these diapirs (Fig. 3), we argue that the source of the rising material is below the detachment; thus, a possible serpentinite composition for the diapirs was investigated (Fig. 4).

Joint interpretation of magnetic and gravity data has been achieved computing a 2.75D forward modeling crossing the diapir in Area 2. The forward model was geologically constrained in depth through seismic data. The 2.75D forward model was approached redrawing the geometry of sedimentary sequences directly in TWT sections. Time–depth conversion was achieved assuming reliable seismic velocities for each block (Fig. 4). The best fit (5.48 nT and 1.354 mGal for magnetic and gravity errors, respectively) was obtained modeling the intrusion as a serpentine diapir with low magnetization (0.015 SI) and density 2.7 g cm$^{-3}$ [24]. Magnetic modeling was limited in depth considering a thermal boundary approximately at 16–17 km estimated assuming a linear increase along a normal thermal gradient.

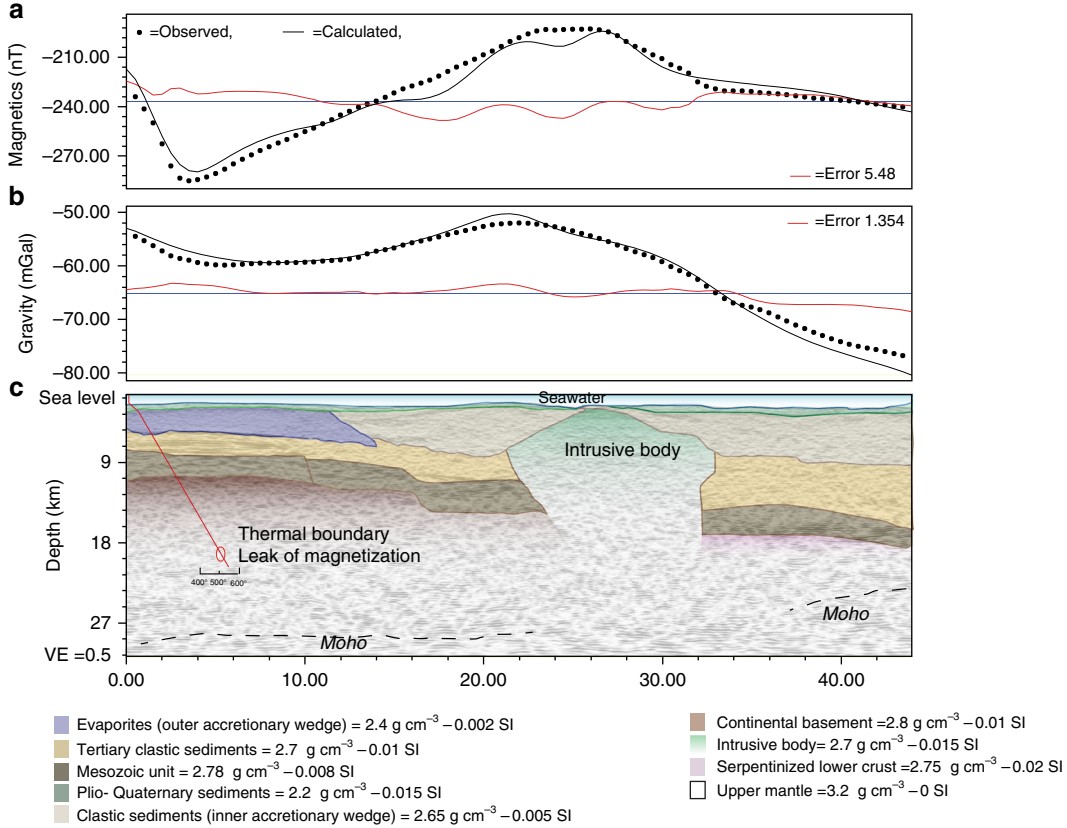

**Fig. 4** Forward magnetic and gravity modeling. **a** Observed, calculated, and misfit (error) curves for magnetic modeling along seismic line CA-A (Fig. 3b). **b** Observed, calculated, and misfit (error) curves for gravimetric modeling along seismic line CA-A (Fig. 3b). **c** Geometry of causative rocks derived from depth-migrated seismic section used as background. Average fitting errors were 5.48 nT (5.5 %) and 1.354 mGal (2.5 %), for magnetic and gravity profiles, respectively. Red line identifies the depth distribution of the local geothermal gradient

Combined P- and S-wave velocity profiles are required to assess whether the seismic structure derived from deep penetration multichannel seismic and potential field data are compatible with a variable abundance of mafic and mantle-derived serpentinite[25]. P- and S- wave profiles are available in the southern region of the diapiric field[26]. Simultaneous inversion of P- and S-wave arrival times, collected during a 3-year ocean bottom seismometer/hydrophone (OBS/H) monitoring campaign (Fig. 5 green stars), yields one-dimensional (1D) P- and S-wave velocity models for the Ionian lithosphere. The 1D model indicates the presence in the Ionian upper mantle of two layers with high seismic P-wave velocity, low S-wave velocity, and high Vp/Vs values, interpreted as partly serpentinized peridotites with 55–65% and 15–25% serpentinization, respectively[26]. These values of Vp/Vs for serpentinites are in agreement with data in the literature[25] and the degree of serpentinization is similar to that derived from gravity and magnetic data. Serpentinization of peridotites produces an increase in susceptibility, because magnetite forms during serpentinization by the breakdown of the iron-rich olivines[27,28]. Serpentinization is also responsible for a density decrease, which follows a linear trend related to the degree of alteration[27]. Considering that serpentinization modifies the physical parameter of the original peridotites, we use our forward model to deduce degrees of serpentinization. According to our model the density of the intruded body is of the order of 2.7 g cm$^{-3}$ with susceptibility of 0.015 SI, suggesting a serpentinization of about 75%[29] and 68%[28], respectively. These values, obtained by two independent data sets, are similar to those

obtained from Vp/Vs data[26], supporting our hypothesis of serpentinite intrusions below the Ionian seafloor.

Comparing these results with our pre-stack depth migrated multichannel seismic data in the same region we find a good correlation between Vp/Vs-derived serpentinite layers and our interpretation of the Ionian lithosphere. In particular, the two serpentinite layers derived from the Vp/Vs data fit well with the source area of the diapirs (Fig. 6b). The geometry and rooting in the Moho of the diapirs combined with the two high Vp/Vs layers at the same source depth support a serpentinite origin for the rising material.

**Pore water geochemistry.** Pore water data from mud volcanoes and other fluid dewatering features are a common tool to constrain subsurface geochemical processes[30–32]. To check for potential geochemical anomalies that may be related to the advection of deep-source fluids, pore waters from gravity cores were analyzed from above the diapiric structure and compared with a reference site (core d) outside this range (Fig. 7 and Supplementary Figs. 9–12).

Core a and core c from above the diapiric structures show deviations from normal diagenetic trends. Specifically, depletion in elements such as Cl behaving conservatively under normal diagenetic conditions, suggests freshening of pore water, e.g., caused by (clay) mineral dehydration. This freshening is typically associated with enrichment of fluid mobile elements, such as Li, B, and Sr[31,32]. Core c shows a significant (10%) depletion in chlorinity and a decrease in $SO_4$ and Ca, both indicating that a major horizon of anaerobic oxidation of methane is present a few

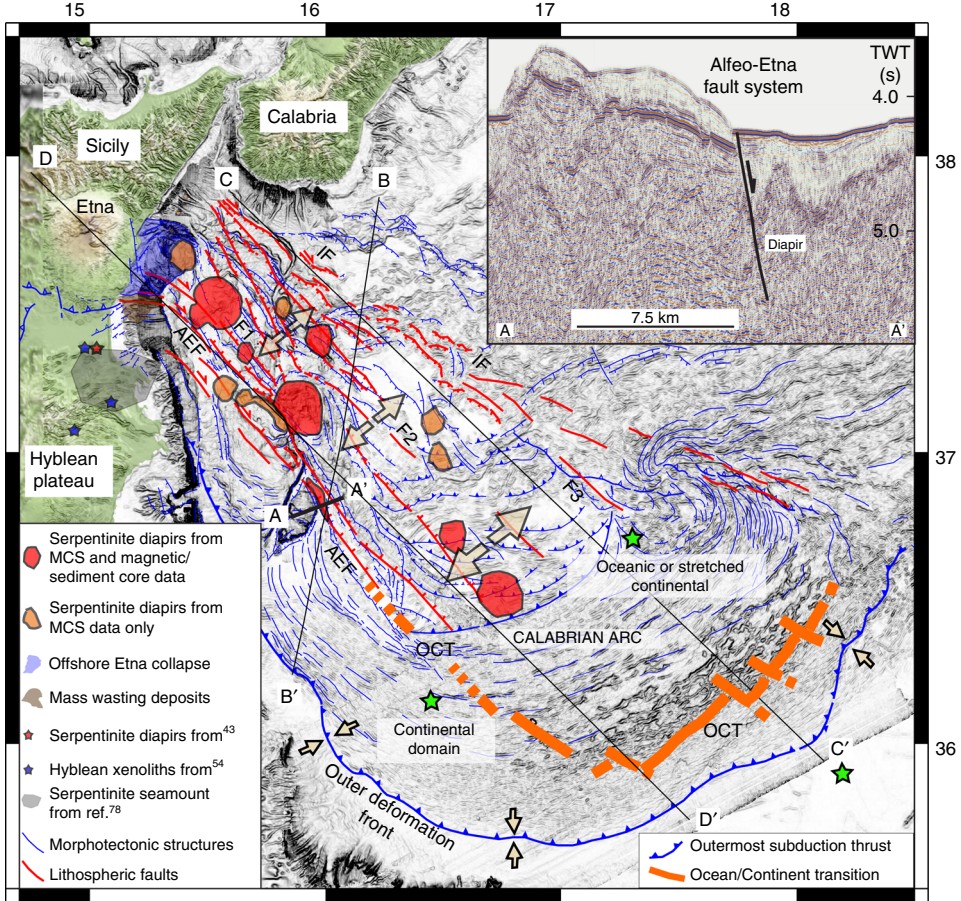

**Fig. 5** Structural map of the Western Calabrian Arc subduction complex with distribution of tectonically controlled serpentinite diapirs. Faulting associated with arc-perpendicular extensional tectonics in the accretionary wedge (modified from ref. [17]) provides an opportunity for fluids to be introduced deep into the underplating plate. A-A': MCS line CALA-02 across a diapir in the sedimentary basin formed along the Alfeo-Etna fault. A diapiric intrusion of serpentinite bearing clayey material in the Hyblean plateau suggests affinity of clays with hydrothermally modified mafic and ultramafic rocks forming the Hyblean lower crust[43,78]. Sections B-B' and C-C' are shown in Fig. 6, section D-D' in Fig. 8. Green stars: OBS/H stations used for P- and S-wave velocity model for the Ionian lithosphere[26]. AEF, Alfeo-Etna fault; IF, Ionian fault; OCT, Oceanic-Continent transition. F1, F2, F3, transtensional faults belonging to the AEF-IF deformation zone[17]

meters below the seafloor. In addition, core-a and core-c show enrichment in dissolved Sr above background values. However, as Sr-enrichment is not accompanied by a freshening trend in both cores and B and Li are not enriched, other processes such as Sr-release from recrystallized carbonates[32] can be suggested. Similar geochemical anomalies have been found in fluids emanating from serpentinite mud volcanoes in the Mariana subduction zone, where slab-derived fluids drive serpentinization and mud volcanism[33,34]. However, serpentinization does not produce a direct typical geochemical signature in sedimentary environments, given variables as composition of the source fluid, nature of ambient sediments, reaction temperatures, etc., causing a wide range of potential variations.

In summary, the observed pore water signature indicates active fluid migration related to subsurface dewatering and associated processes. This is confirmed by sediment facies analyses showing fluid flow related features such as patchy cloudy facies[35], sediment layers disruption, mud injections, and vertical fluid paths (Supplementary Fig. 10).

The CA subduction system is close to a continental collision that causes margin segmentation and disruption of the subduction system driven by pre-existing heterogeneities in the forearc basement[17]. Crustal velocity models obtained from deep seismic sounding (DSS) data[12] account for a Moho depth decreasing from

about 35 km in the EL, to some 20 km in the WL, close to the Malta escarpment. Moho depth variations occur in the region between the AEF and IF, where our seismic reflection profiles (Fig. 3) show a Moho shallower than in the surrounding regions, in agreement with other evidence, suggesting a depth between 12 and 15 km[10]. This geometry suggests an asymmetric rifting affecting a large corridor between the AEF and IF, and the collapse of the WL of the accretionary wedge. Thus, the AEF may represent the western master fault accommodating plate divergence along a dextral shear zone, triggering serpentinite diapirism.

Differences in plumbing systems and hydrogeological characteristics of forearc basements may trigger contrasting processes in different forearcs, including fluid flow and mud volcanism (i.e., Nankai Trough or Mediterranean Ridge[36]) or serpentinite diapirism (Izu-Bonin forearc[37–39]), whereas impermeable and consolidated sediments tend to inhibit fluid circulation (Japan Trench[40]). The CA, the narrowest subduction system on Earth, hosts both processes: mud volcanism in the EL[35,41,42], serpentinite diapirism in the WL (this study), as well as on the Hyblean Plateau close to the AEF onland[43].

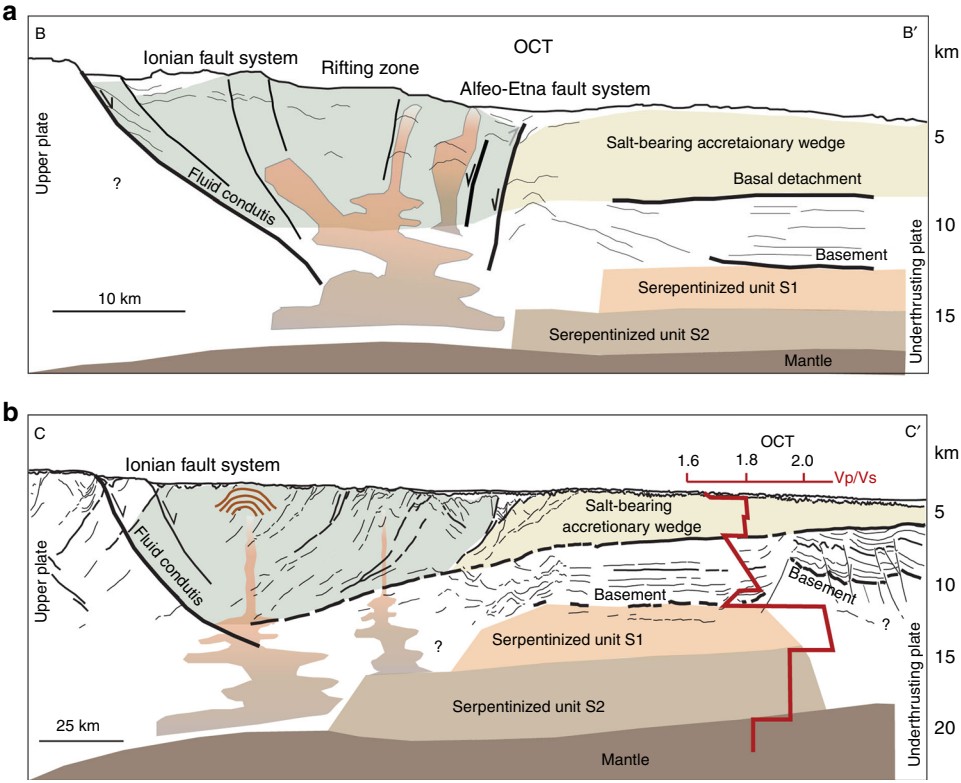

**Fig. 6** Structural cross-sections of the continental margin. **a** Sketch based on MCS line MS-107 showing the margin structure parallel to the arc where lithospheric transtensive faults trigger the rise of serpentinite material from the lower plate. **b** Sketch based on MCS line CROP-M2B showing the margin structure orthogonal to the arc where the geometry of the subduction thrust is imaged. The presence of serpentinite units and Vp/Vs data in the outer part of the subduction complex is taken from[26]. OCT, Oceanic-Continent transition. Location of seismic sections in Fig. 5

## Discussion

Mantle-derived serpentinites are found along slow to ultraslow spreading ridges, at magma-poor rifted margins and at subduction zones, where the downgoing plate may trigger arc magmatism or hydrate the deep Earth[44–46]. We suggest three possible scenarios for serpentinite emplacement in the Ionian Sea: during Pleistocene shear processes and fluid migration along the AEF; as a consequence of CA subduction; or during the formation of the Tethyan Ocean.

Diapirs alignment along the AEF suggests a direct link between tectonic activity and uprising of low-density material. However, we exclude that serpentinites formed in Pleistocene times since serpentinization is not limited to the fault zone, as indicated by velocity models of the lithosphere[26] (Fig. 6). Tectonic dilatation in the rifting zone of the western Ionian Sea may trigger downwards fluid flow driving mobilization of serpentinites inherited from the Mesozoic ocean (Figs. 6 and 8) and already emplaced in the underplating African plate.

A different scenario for serpentinite emplacement includes the mantle wedge at the subduction zone, possibly hydrated by the subducting plate. However, this scenario is unlikely in our region, because of the external position of the diapirs not far from the African foreland, in an area where the slab is rather shallow (Fig. 5). Velocity models for the Ionian lithosphere[26], suggest two upper mantle layers (3.3 and 5 km thick, respectively, at 8 and 16 km depth), with high Vp/Vs ratio (1.95–2.1), interpreted as due to partly serpentinized peridotites[26]. The high Vp/Vs ratio suggests low pressure-low temperature serpentinization[47], different from that inferred usually for subduction settings. Serpentinites may thus characterize the basement of the southern Ionian Sea in a region as wide as hundreds km$^2$ at the transition with the African foreland.

If serpentinites are inherited from the Tethyan ocean, they are Mesozoic in age, and could have developed along fractures located near the boundary between the oceanic crust and the adjacent thinned continental crust, a situation similar to that observed west of the Galicia Bank along the Iberian margin[24,48,49]. Serpentinization of peridotites is due usually to hydrothermal processes and fluid circulation associated with early stages of accretion during rifting. As the sub- Ionian Sea crust has never been sampled and its nature never unambiguously defined, we cannot exclude that this part of the Ionian basin formed after tectonic extension and exhumation of a continental mantle causing serpentinization and fluid flow. In magma-poor margins, crustal faulting and mantle serpentinization are believed to occur when the crust becomes entirely brittle after a certain amount of extension[50]. In both scenarios, the AEF would represent a major Mesozoic tectonic discontinuity. Formation of serpentinites, regardless of geodynamic setting, is thus a consequence of lithospheric mantle exhumation as a result of a bulk pure shear, or related to Mesozoic rifting that brought the Moho and upper mantle rocks close to the surface. In any case, serpentinization could take place if suitable fluid pathways existed, with faults cutting down from the seafloor to the mantle[44]; the AEF may represent one such major discontinuity.

Interesting inferences can be obtained from xenoliths sampled from basalts and diatremes of the Hyblean Plateau (SE Sicily) East of the study region (Fig. 5). The Hyblean Plateau has been considered traditionally an uplifted portion of the Africa continental plate[51,52]. However, among the abundant lower crustal/upper mantle Hyblean xenoliths, none are of sub-continental derivation[53]. Hyblean xenoliths include serpentinized harzburgites with a relict primary spinel Cr number between 0.25 and 0.30; this and other geochemical indicators are within the range of oceanic

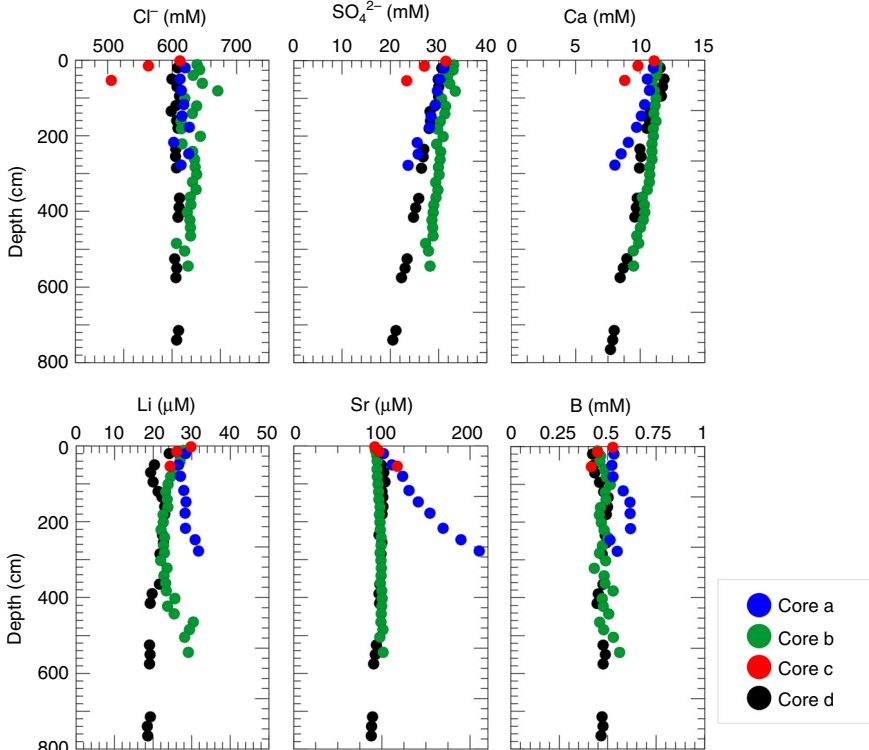

**Fig. 7** Depth profiles of selected pore water constituents in the sediment cores. Core a: SF17-04; core b: GH15-03; core c: SF17-05; core d: CQ14-02 (reference core not affected by diapirism). Core a and c from above the diapir show deviations from normal diagenetic trends (core d). In both cores, pore water Sr concentrations increase continuously with depth; in core a, this trend is accompanied by depletion in Cl (and other elements that typically behave conservative at this depth below the seafloor). These geochemical anomalies provide evidence for fluid advection (affected probably by mineral dewatering and other deep-seated geochemical processes) and suggest a direct relation to the observed diapirism. Core location in Fig. 2

peridotites[53], suggesting that the lower crust may include bodies derived from an ancient oceanic domain[54]. Some Hyblean xenoliths consist of metasomatized gabbroic rocks of lower crustal origin probably derived from an oceanic or OCT environment[54]. Hydrothermal modification of xenoliths suggests that serpentinite-hosted hydrothermal systems[54], typical of slow and ultra-slow spreading ridges, were active since the Middle Triassic, as deduced from U/Pb ages of hydrothermal zircons[55]. These results, backed by geophysics[56,57], suggest a serpentinized oceanic-type lithosphere beneath the Hyblean Plateau and may imply some connection between sub Ionian Sea and sub Hyblean lithosphere.

Based on the P-wave velocity model proposed for southeastern Sicily[56], petrological evidence for the lithosphere beneath the Hyblean Plateau and the occurrence of serpentinized harzburgite xenoliths in Hyblean diatremes were used to estimate the degree of serpentinization[53]. The resulting values are in line with the presence of peridotites affected by different degrees of serpentinization (35–100 vol.%) at 8–19 km depth, with the Moho discontinuity as a serpentinization front at about 19 km[53] in agreement with results from the Ionian Sea[26], further stressing a connection between the Ionian and Hyblean lithosphere.

Serpentinite diapirs below the Ionian Sea would have important implications to reconstruct the geometry and processes that have driven Tethyan ocean rifting. Our reconstruction, pointing towards mantle serpentinization coeval with Tethyan rifting, should be considered in the frame of Mesozoic paleogeography. Widespread serpentinites suggest that the Tethyan Ocean, at least in this domain, was not a magma-rich basin; serpentinization may have occurred along the OCT or may represent the product of hydration of exhumed mantle. Accordingly, the AEF may represent one of the shear zones where extension was focused,

similar to the Magnaghi and Vavilov sub-basins of the Tyrrhenian Sea, where tensional faults favor mantle exhumation and percolation of water in the deep crust[58]. Alternatively, if the Tethyan basin was floored by oceanic crust, the AEF might represent the OCT where the crust was thin enough to become entirely brittle during rifting, as in the Porcupine Basin and Galicia margin[24,44,49]. Gravity data (Supplementary Fig. 13) provide evidence of OCT geometry. We suggest that gravity highs marking the AEF trend and basement topography below the accretionary wedge[16] (Figs. 2 and 6) represent the transition between continental and oceanic or continental stretched domains (Fig. 5 and Supplementary Fig. 13). The complex geometry of the OCT reflects the 90° curvature of the continental margin close to the Sirte gulf (Fig. 1). In this reconstruction, oceanic crust, if present, should be limited to the inner accretionary wedge of the subduction complex and the AEF should be located close to the Mesozoic Tethyan OCT (Fig. 5).

Our reconstruction suggests that diapirism of inherited serpentinites occurs along active transverse shear zones segmenting the CA subduction complex, which thus controls structural development and fluid flow. Mt. Etna volcano is aligned with the serpentinite diapirs; its volcanism appears to be linked to active extensional tectonics, because the volcano is located on the NW tip of the AEF and has the same age of the inception of transtensional deformation in this sector[17]. Although Multi-Channel Seismic (MCS) data suggest that both the AEF and IF develop over structural boundaries inherited by the Mesozoic Tethyan basin, their recent tectonic activity fits with the latest Quaternary extensional phase. In fact, fault inception along the AEF produced the down throw of the WL and the formation of sedimentary basins, filled by up to 700–800 m-thick sediments (Fig. 3b s.p.

11,200). This observation was used to reconstruct the age of the fault assuming a sedimentation rate derived from sediment cores collected in the region[18,59], indicating that 700 m-thick sediment along the AEF might correspond to 350,000–700,000 years, in agreement with the age of Mt. Etna volcano[60]. This connection between large-scale offshore tectonic processes and the formation

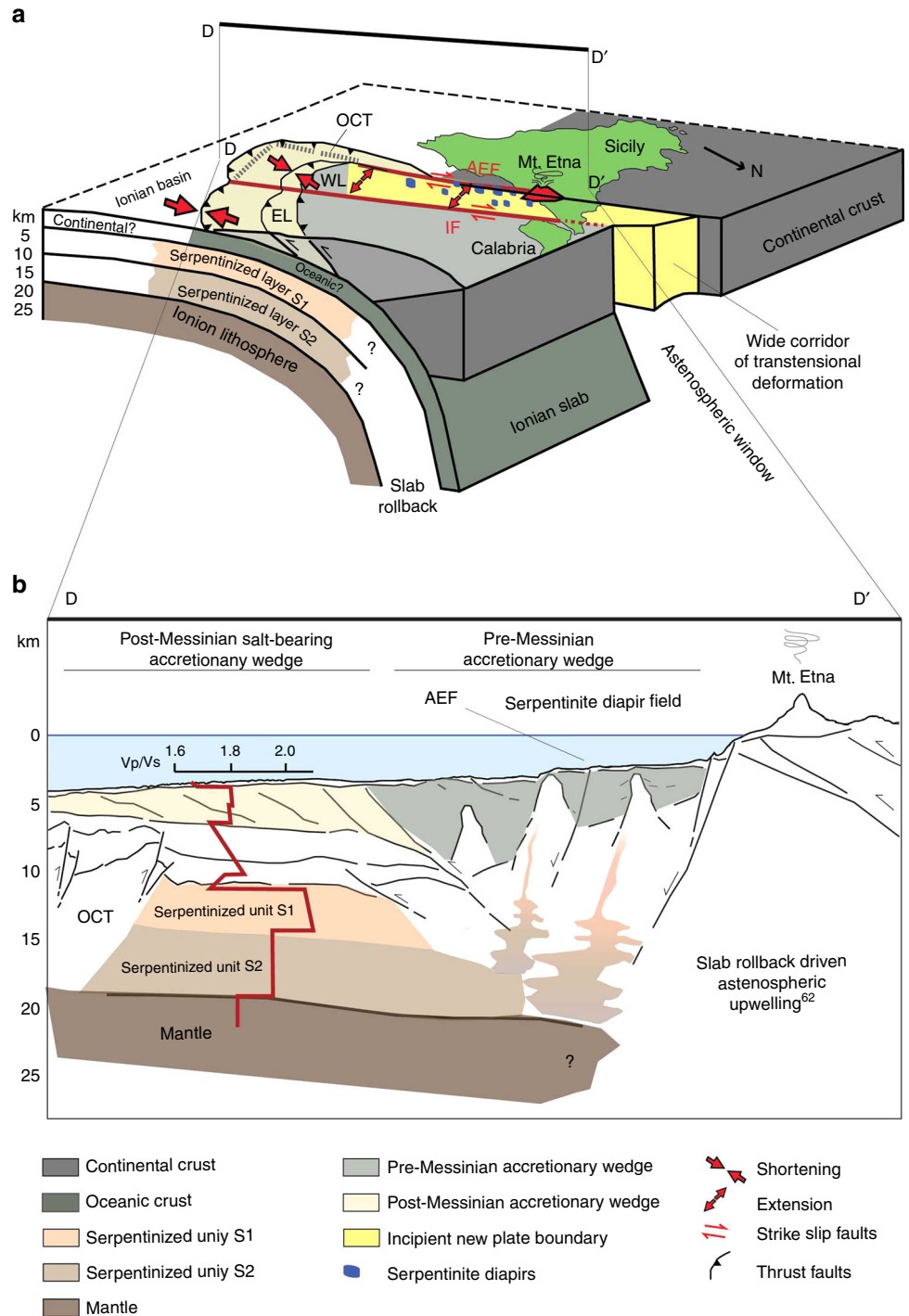

**Fig. 8** Schematic block diagram and cross section of the Western Calabrian Arc. **a** Block diagram of the Western Calabrian Arc subduction complex showing relationships between deep slab, transtensional faults, serpentinite diapirs, and Mt. Etna volcano (modified from refs. [62,69]). Faulting associated with arc-perpendicular extensional tectonics in the accretionary wedge (modified from ref. [17]) is shown in the yellow corridor between the Ionian and Alfeo-Etna fault systems. Extensional processes and associated Mt. Etna volcanism could be related to vertical upwelling of the asthenosphere at the SW lateral edge of the Ionian slab[62,63,79]. At the edges of a retreating slab the toroidal component of asthenospheric flow may induce an upward flow and decompression melting which could explain the formation of Mt. Etna[2]. Transtensional processes along the rifting zone between the Alfeo-Etna and Ionian fault may drive inherited serpentinite diapirism during slab tearing processes. **b** Sketch of the margin structure orthogonal to the arc based on interpretation of available multichannel seismic and Vp/Vs data[26]. Section D-D' (location also indicated in Fig. 5) crosses the Calabrian Arc accretionary wedge, the inherited serpentinite diapir field, and Mt. Etna implying structural control on both diapirism and Mt. Etna formation. AEF, Alfeo-Etna fault; EL, eastern lobe of the accretionary wedge; IF, Ionian fault; OCT, Oceanic-continental transition; WL, western lobe of the accretionary wedge

**Table 1 Acquisition parameters, processing sequence and resolution of geophysical data used in this study**

| Geophysical multi-scale data set | Acquisition parameters | Main processing sequence | Resolution in space |
|---|---|---|---|
| CROP MCS data set | Source: 4,906 cubic inch air guns<br><br>Streamer: 4,500 m<br>Group interval: 25 m<br>Shot interval: 62.5 m<br>Coverage: 3,600%<br>Sampl. interval: 4 ms | Full pre-stack depth-migration (PSDM), with SIRIUS/GXT, Migpack software package. | km-scale |
| ETNASEIS data set | Source: air guns array<br>Streamer: 2,400 m<br>Coverage: 2,400% | Velocity analysis, stack, time migration | km-scale |
| Mediterranean Sea (MS) MCS data set | Source: Flexotir (2 guns)<br><br>Streamer: 2,400 m<br>Group interval: 50 m<br>Shot interval: 100 m<br>Coverage: 1,200%<br>Sampl. interval: 2 ms | Velocity analysys, stack, DMO, velocity analysis, stack, and migration | km-scale |
| ENI MCS lines CA-A, CA-B and CA-C | Source: 3,400 cu inch air guns<br><br>Streamer: 6,200 m<br>Group interval: 12.5 m<br>Shot interval: 37.5 m<br>Coverage: 12,000%<br>Sampl. interval: 2 ms | Velocity analysis, stack, migration | $n \times 100$ to km-scale |
| CALAMARE MCS data set | Source: 2 Sodera G.I. guns,<br><br>Streamer: 600 m Group int.: 12.5 m<br>Shot interval: 50 m<br>Coverage: 600%<br>Sampl. interval: 1 ms | Velocity analysis, stack, DMO, velocity analysis, stack, migration | $\times 100$ m to km-scale |
| Sparker seismic data | Source: 30 kJ Teledyne system<br><br>Streamer: active section: 50 m, single channel<br>Shot interval: 4–8 s (12–24 m) | | nx100 m to km scale |
| Chirp data set | 17 hull mounted 17 transducers CHIRP-Benthos sonar system (3–7 KHz sweep frequency) | Data represented through variable density sections with instantaneous amplitude | Metric/decimetric |

of the volcano appears thus to be supported also by the age of transtensive reactivation along the AEF, which might indicate a primary role played in Mt. Etna volcanism by a Pleistocene geodynamic re-organization.

Etna Na-alkaline geochemistry[61] does not indicate a deep slab magma source, as in the Aeolian K-alkaline magmatism of the southern Tyrrhenian. It may be related to a subduction rollback driven vertical 'slab window' that might explain its alkaline chemistry and the anomalous external position of the volcano, not derived from the mantle wedge but in some way related to the slab[2,62,63]. Moreover, recent Etnean basalts composition may reflect addition of fluids to the mantle source and/or selective crustal contamination[12] occurring when the AEF shifted from compression with extension during Pleistocene[17]. This agrees with the voluminous melting under Mount Etna having originated from below the African plate[62]. Although it cannot be excluded, there is no geochemical evidence for a direct role of serpentinites in the genesis of Mt. Etna volcanism. Boron isotopes of Mt. Etna lavas do not support a role of serpentinite-derived fluids in their genesis[64]. However, a major tectonic discontinuity has been invoked as leading to the formation of Mt. Etna[17,19,64]. This fault would represent a window allowing upwelling of asthenosphere from below the subduction system between the Ionian and Sicilian slabs[62] (Fig. 8). On the other hand, shallow partial melting as a response to mantle decompression was suggested as a source of Mt. Etna magmas within the framework of

extensional tectonics[65]. Both scenarios (i.e., deep and shallow source) call for a major discontinuity, i.e., the AEF. Fluid addition along the fault coupled with a pressure drop might induce generation of magmas with the peculiar geochemistry of Etnean lavas[64]. The same process along the wide corridor between the Ionian and AEFs might have triggered offshore inherited serpentinite diapirism.

Serpentinite diapirs in the Ionian Sea provide evidence on how altered mantle rocks may be transferred from rifting, to basin floor, to subduction[66] where they can give rise to large scale diapirism, enhancing shear processes and margin disruption during the final closure of the ocean. In this reconstruction, lithospheric inheritance controls neotectonic activity and location/reactivation of tectonic structures, as also observed in Canada[67] or in the Eastern Mediterranean[68]. The transition from weak and stretched continental crust to the relatively strong oceanic crust appears to guide present day seismicity patterns as well as plate kinematic evolution of this region. It was suggested that serpentinites control the seismic or aseismic behavior of subduction zones[38]. Major historical earthquakes in the CA (i.e., the Messina 1908, Catania 1693, and Sicily 1169 earthquakes) were generated in an area close to the Ionian and AEFs[18,69,70]. Moreover, the distribution of seismicity and the lack of thrust-type earthquakes along the CA subduction complex in the Ionian Sea may be due to sub Ionian serpentinites influencing the mechanical coupling between plates as proposed for the Japanese

margin[71]. Low thrust-type seismic activity may be an indication that subduction and convergence is aseismic, or that plate motion is expressed by large earthquakes with long recurrence time intervals. Serpentinites, when thicker than 2–3 km, may form an efficient flow channel at the top of the subducting slab influencing seismogenesis[72]; this has to be taken into account for seismic hazard assessments in this densely populated region.

The discovery of serpentinite diapirs in the CA provides the first example of a new class of serpentinites in a subduction system derived directly from the lower plate through lithospheric extensional faults disrupting the accretionary wedge and driven by tectonic processes at the slab edge. These findings may lead towards a more complete understanding of the structure of the Ionian lithosphere, of the mechanisms exposing altered products of inherited mantle peridotite at the seafloor long time after their formation, and finally of the role of serpentinization in driving continental collision.

## Methods

**Multibeam data.** The morphobathymetric map is derived from three datasets collected by ISMAR-CNR during different cruises onboard of *R/V CNR Urania*, integrated by the 500 m grid DTM from CIESM/IFREMER Medimap group[73]. Swath bathymetry during the *Urania* cruises (2010, 2011, and 2015) was collected using a SIMRAD EM 302 Multibeam Echosounder. During those cruises, a keel-mounted probe provided real-time estimate for sound-velocity at the surface, whereas sound velocity profiles were performed daily to correct for density, salinity, and temperature variations in the water columns. Functioning of the multibeam echosounder was controlled by the Kongsberg software SIS, which provided data quality control and information for the survey plan. Multibeam data were subsequently processed using Caris Hips and Sips suite software.

**Seismic reflection data.** A close-spaced grid of seismic reflection profiles, characterized by different penetrations and vertical resolutions, was used to define the structural setting of the CA subduction complex in the Ionian Sea (Supplementary Fig. 1a). The deep structure was reconstructed from interpretation of MCS reflection profiles collected during different cruises (ENI-CA, Etnaseis, CALA-MARE and Mediterranean Sea—MS data sets), from the 70s to the 90s. Single-channel Sparker profiles (the J data set of ISMAR-CNR) were used for detailing shallow structure of key areas previously identified through the analyses of deep penetrating seismic data. Acquisition and processing parameters of each set of data are described in Table 1.

Final data processing and visualization, as well as structural mapping was performed using SeisPrho software[74], freely distributed at http://software.bo.ismar.cnr.it/seisprho.

We used all available seismic data (CNR_ENI Deep Crust Seismic Profiles—CROP, ENI CA MCS seismic, MCS CALAMARE lines, Sparker J data set, and Chirp profiles), characterized by different vertical resolutions (Table 1), to describe geometry and kinematics of the fault systems segmenting the western Ionian Sea.

**Magnetic and gravimetric data.** Shipborne magnetic data were acquired during CALAMARE08 cruise[16] using a SeaSpy Marine Magnetics magnetometer, towed 180 m astern of the vessel. Raw magnetic data were collected at 1 Hz sampling frequency, resulting in an average spacing between every single measurement of about 4 meters.

Positioning of the magnetic sensor was obtained by a FUGRO differential Global Positioning System (GPS) mounted onboard of the *R/V Urania* after performing the layback correction. Despiking and low-pass filtering were applied to the raw data to attenuate high frequency noise. Magnetic anomalies were obtained removing the International Geomagnetic Reference Field (IGRF) geomagnetic reference field. Final magnetic anomalies (Fig. 2) were gridded using a minimum curvature method at 500 m grid cell size.

Gravimetric profiles were acquired along each CALAMARE08 seismic profile track using a LaCoste and Romberg AirSea Gravity System II at 1 Hz of sampling frequency. Gravity values were smoothed at 120 s by a low-pass filter, with an Exact Blackman Window. Drifting of the gravimeter was estimated at the beginning and end of the cruise; the average drift was below 0.2 mGal per day. Accuracy in positioning and velocity measures guaranteed 1 mGal accuracy in the gravity anomaly. Gravity meter readings were tied to an absolute gravity station in Naples, by using a portable model G gravity meter. Drift, Eötvös and Latitude corrections were applied to obtain the Free Air anomaly.

Quantitative interpretation of crustal properties below the CA-A seismic line track was obtained performing a 2.75D forward modeling, which allows geometry and density/susceptibility values of crustal bodies in strike direction to be defined. Magnetic and gravity data were sampled along the CA-A track using 400 data points (about 10 sample per km). The model was constrained by geometries and crustal features of major crustal bodies determined by seismostratigraphic interpretation of depth-migrated seismic sections. Seismic velocity of each stratigraphic interval was converted into density using Gardner's equation[75], which produced a preliminary set of values for subsequent re-adjustments. Best-fit between gravity and magnetic modeling was achieved by fine-tuning susceptibility and density values. After fixing the geometry of the diapiric body and the properties of the overburden, we tested the effect of different lithologies, including fine-grained clastic sediments, gabbroic cumulates, and a typical oceanic crust of slow-spreading domains. We finally observed that all realistic lithologies, alternative to that of serpentinized peridotites, provided different gravity and magnetic patterns relative to those observed in the field (Supplementary Fig. 8).

**Pore water analyses.** Pore water samples were extracted using Rhizones[76] from cooled cores. Concentrations of chloride and sulphate were measured by ion chromatography (METROHM 761). Acidified subsamples were analyzed by inductively coupled plasma optical emission spectrometry (VARIAN 720-ES). Details on analytical methods applied routinely at GEOMAR are available at http://www.geomar.de or in publications[31].

**X-ray diffraction analysis of core GH15–03.** The homogenized sediment sample A1 (Supplementary Figs. 10 and 11) was split and air-dried, and glycolized sub-samples were analyzed by X-ray diffraction (Cu Kα, 3–70°, 0.02° steps). Mineral components and quantitative estimates were determined from the diffraction patterns, by using the software package XPowder.

**Satellite-derived vertical gravity gradient.** Vertical Gravity Gradient (VGG) data are from a global marine gravity model compilation[77]. CryoSat-2 and Jason-1 satellite missions provided high track coverage that enhanced the gravity anomaly signal within 12–40 wavelength band and a maximum resolution of 2 mGal. VGG computation permits to remove long wavelengths of gravity field particularly related to large-scale Moho-isostatic effects[77]. Peculiar tectonic structures and specific seafloor fabric are visible in the VGG data, which highlight density variation at a frequency band typical of upper/middle crust. The VGG model computed starting from satellite gravity data permits to distinguish OCT's geometry[68], as well as relict ridge structures below thick crust[77]

**Data availability.** Data associated with this paper are available upon request (alina.polonia@ismar.cnr.it).

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

## Acknowledgements

We are greatly indebted to A.L. Cazzola, A. Fattorini, and C. Cattaneo (ENI Spa) for providing us with MCS lines CA-A, CA-B, and CA-C, to R. Nicolich for MCS line ETNASEIS-6, to OGS (Istituto nazionale di oceanografia e di geofisica sperimentale, Trieste) for MCS lines MS-26 and MS-107; M. Ligi for fruitful discussions and acquisition of seismic reflection line CALA-02. Cores "a" and "c" were collected during the "Seismofaults 2017" cruise onboard of R/V CNR Minerva-Uno (A. Billi, M. Cuffaro, and the Seismofaults 2017 team). We thank B. Domeyer, A. Bleyer, and R. Surberg for pore water analyses, Jutta Heinze for assisting XRD measurements, and Katja Lindhorst for pore water collection in core CQ14-02. We thank C. J. Garrido and an anonymous reviewer for their detailed and constructive reviews, which have contributed to improve the final quality of the manuscript. This work was supported by the project FLOWS (EU-COST action ES 1301). Morphobathymetric and structural maps were generated using the GMT software[80]. We wish to dedicate this work to the memory of Giovanni Bortoluzzi who suddenly died in September 2015. The acquisition of geophysical data in the Ionian Sea was possible thanks to his valuable contribution. This is ISMAR paper n. 1944.

## Author contributions

A.P. conceived and designed the research. A.P., L.T., L.G., and S.R. acquired data at sea, and completed data processing and interpretation. L.C. and F.M. acquired potential field at sea and completed potential data processing and interpretation. E.B., C.H., and M.S. performed the mineralogical observations, PW analyses, and integrated data interpretation. A.A. and M.C. performed structural interpretations on seismic data. All authors discussed the problem, methods, analyses and results, and reviewed the manuscript.
