## [Peer Review File · Nature Communications]

Reviewers' comments:

Reviewer #1 (Remarks to the Author):

This paper presents new multibeam, and multichannel seismic reflection, magnetic and gravity field data and core fluid geochemistry to support the presence of mantle-derived serpentinites deriving from the lower plate in the external subduction system of the Calabrian Arc. If it had been more convincingly demonstrated the presence of mantle-derived serpentinite related to subduction processes, this would be, to my best knowledge, one of the first report of the presence of subduction-derived serpentinite in the lower plate of a subduction system. Although the existence of serpentinites in the lower plate it is an interesting discovery, it is unclear what the impact of this finding would be in the wider field of the dynamics of subduction zones, or, alternatively, in our understanding of the mechanisms of mantle exhumation in OCT zones. The main strength of the paper is the unique multibeam, seismic reflection dataset of the Calabrian Arc, which provides very detailed imaging of the structure of this complex Mediterranean subduction system. The main weaknesses of the paper are three:

(i) Evidence of lower plate mantle diapirism: The paper present not evidence strongly supporting the existence of serpentinite diapirs, although I agree that geophysical data may well be interpreted as serpentinite diapirs. Unfortunately, as concluded by the authors, the fluid geochemistry evidence presented in the paper is not conclusive of active or past mantle serpentinitization. Lacking such evidence, the authors claim the existence of mantle-derived serpentinite based on gravity and magnetic modeling; however, this modelling is very dependent on the forward modelling approach and parameters, and usually does not provide firm evidence to discriminate mantle-derived serpentinite from other lithologies (e.g., Escartin & Cannat, 1999; Dunn et al., 2007; and references therein). This is well known in slow spreading mid-ocean ridges where, to date, gravity and magnetic modeling has been unable to provide solid evidence to unambiguously discriminate serpentinitization at depth from other combinations of mafic and ultramafic rocks (e.g., Escartin & Cannat, 1999; Dunn et al., 2007; and references therein). Gravity and magnetic modeling can also be consistent with magmatic cumulates, making it difficult to distinguishing between gabbro and serpentinitized peridotite (e.g., Escartin & Cannat, 1999; Dunn et al., 2007; and references therein). Combined P- and S-wave velocity profiles would have been necessary to assess if the seismic structure is compatible with variable abundance of mafic and mantle-derived serpentinite (e.g., Carlson and Miller, 1997). I agree, though, that the geometry and apparent rooting in the Moho, supports serpentinitization, but does not exclude they can be mafic to ultramafic magmatism intrusions.

(ii) The second weakness of the paper is that the age putative serpentinite diapirs, their genesis, and their relation to Calabrian subduction zone is not well constrained and highly speculative. As pointed out by the authors, the shallow depth and their plate location make it unlikely that such putative serpentinite diapirs are related to Calabrian subduction processes. They are most likely inherited from Tethyan ocean formation (i.e, Mesozoic in age), as the authors conclude in the highly speculative section "Origin of serpentinites and the Mesozoic Tethyan basin". If they are inherited Mesozoic features, the title and the main contribution of the paper are rather misleading because there is not relationship between "Lower Plate Mantle Diapirism" and "Convergent margin" processes, a relationship one might expect to be the main contribution of this report from its title. Most of this section speculates about the formation of such diapirs during Mesozoic rifting of the Tethys Ocean or in OCT in a non-convergent setting. The authors should refer to it as "Inherited, Lower Plate Serpentinite Diapirism" unrelated to the Convergent margin processes. On the other

hand, the term “Mantle Diapirism” is unfortunate: at best, it is Serpentinite Diapirism. Mantle Diapirism refers to ductile upwelling —usually associated to decompression melting— of mantle-derived peridotite. Serpentinite diapirs are produced by buoyant upwelling or tectonic exposure of serpentinites formed by hydrothermal alteration of non-mantle- or mantle-derived rocks. The exact mechanisms and age of formation of such diapirs, and their relevance for Calabrian subduction system and for subduction systems overall are not compellingly demonstrated in the manuscript. Throughout the manuscript, the authors overstress the importance of the serpentinites, without really going into the details of the process. For instance:

(lines 166-171) “The newly discovered serpentinite diapirs field below the Ionian Sea have important implications. First, diapirs may carry to the surface information about the nature of the crust presently still controversial, and about the geometry and processes that have driven Tethyan ocean rifting”.

Even if they carry such information, how this information is going to be retrieved to understand the nature of the crust? How the geometry and processes that have driven Tethyan ocean rifting are going to be studied? A non-lower plate exposure in a divergent setting (OCT, or back-arc basin) is a much better location to investigate such processes because they will not be masked by the upper plate. Better constraints can be obtained, for instance, from the nearby Tyrrhenian basin, which exposes mantle peridotites and it is undisturbed by subduction (Bonatti, 1999).

(lines 199-201) “Serpentinite diapirs in the Ionian Sea provide evidence on how mantle rocks may be transferred from rifting, to basin floor to subduction where they can give rise to ophiolites, generally regarded as fragments of subducted oceanic crust emplaced onto continental crust through underplating and tectonic erosion”.

It is uncertain how the evidence provided in the manuscript from the Calabria subduction system can help to understand the origin of supra-subduction ophiolites, which usually provide a magmatic record of supra-subduction magmatism, and discriminate from those formed in mid-ocean ridges.

(lines 202-204) “Our Ionian Sea findings show that serpentinites may be already in existence before entering the subduction channel and can reach shallow levels of the subduction complex through large-scale diapirism, enhancing shear processes and margin disruption during the final closure of the ocean.”

The authors has not proven that such diapirism is related to subduction processes; actually fore arc serpentinite diapirism is related to active serpentinization, which is not demonstrated here. This conclusion somewhat contradicts their earlier conclusion on the formation of serpentinite diapirs during Mesozoic rifting in a non-convergent setting.

(iii) The last weakness of the paper is the relationship between the putative serpentine diapirism and recent volcanism at Mt. Etna. This link is highly conjectural, lacks solid geochemical evidence, and the cause-effect between serpentine diapirism and volcanisms it is not demonstrated. The relationships between the reported diapiric alignments towards the Hyblean plateau is something worth to investigate in greater depth, but it is not compelling evidence for the role of serpentine in the genesis of the Mt. Etna volcanism. There is direct evidence (non-cited) that the unexposed lithospheric roots of the Hyblean plateau consist of relict of ultraslow-spreading Mesozoic Ionian-Tethys Ocean exists in the xenolith suite of the Hyblean tuff-breccia deposits (e.g., Scribano et al., 2006; Manuella et al., 2013). Most of Hyblean xenoliths contain evidence of abyssal-type hydrothermal metasomatism, including serpentinization of ultramafic rocks (Scribano et al., 2006; Manuella et al., 2013; Simakov et al., 2015). I think that this evidence, combined with the geophysical imaging provide in this manuscript, is perhaps the stronger evidence of these alignments being serpentinite diapirs and their inheritance from the lower plate. A separate issue is a cause-effect relationship between the presence of such serpentinite diapirs and the origin of the Mt. Etna volcanism. Boron isotope geochemistry provides unequivocal evidence of role of serpentinite-derived fluids in the genesis of volcanism (e.g., Straub and Layne; 2002; Tonarini et al., 2007), but, to my knowledge, boron isotope systematic of Mt. Etna volcanism does not support

such derivation (Tonarini et al., 2001). Even if cannot be excluded, there is not geochemical evidence for the role of serpentinite in the genesis of Mt. Etna volcanism. The authors argue the involvement of serpentinite in the genesis of Etna volcanism, mostly on the basis of a fluid-rich environment and its provenance from the lower plate of this volcanism: (lines 196-198) "When the AEF tectonic behavior shifted from compression to extension, the volcanic products of Mt. Etna reached the surface together with serpentinite diapirs, in agreement with the voluminous melting under Mount Etna having originated from under the African plate."; however, These arguments that do not necessarily evidence the role of serpentinites in the genesis of Mt. Etna volcanism.

REFERENCES:

- Bonatti, E., Seyler, M., Channell, J., Girardeau, J., Mascle, G., 1990. Peridotites drilled from the Tyrrhenian Sea, ODP Leg 107, Peridotites drilled from the Tyrrhenian Sea, ODP Leg 107, pp. 37-47.
- Carlson, R.L., Miller, D.J., 1997. A new assessment of the abundance of serpentinite in the oceanic crust. *Geophysical Research Letters* 24, 457-460. 10.1029/97GL00144.
- Dunn, R.A., Forsyth, D.W., 2007. 1.12 - Crust and Lithospheric Structure – Seismic Structure of Mid-Ocean Ridges A2 - Schubert, Gerald, *Treatise on Geophysics*. Elsevier, Amsterdam, pp. 419-443.
- Escartin, J., Cannat, M., 1999. Ultramafic exposures and the gravity signature of the lithosphere near the Fifteen-Twenty Fracture Zone (Mid-Atlantic Ridge, 14 -16.5 degrees N). *Earth and Planetary Science Letters* 171, 411-424.
- Manuella, F. C., Brancato, A., Carbone, S. & Gresta, S, 2013. A crustal-upper mantle model for southeastern Sicily (Italy) from the integration of petrologic and geophysical data. *J. Geodyn.* 66, 92-102 (2013).
- Straub, S.M., Layne, G.D., 2002. The systematics of boron isotopes in Izu arc front volcanic rocks. *Earth and Planetary Science Letters* 198, 25-39.
- Scribano, V., Ioppolo, S. & Censi, P, 2006. Chlorite/smectite-alkali feldspar metasomatic xenoliths from Hyblean Miocenic diatremes (Sicily, Italy): Evidence for early interaction between hydrothermal brines and ultramafic/mafic rocks at crustal levels. *Ophioliti.* 31, 161-171.
- Simakov, S.K., Kouchi, A., Mel'nik, N.N., Scribano, V., Kimura, Y., Hama, T., Suzuki, N., Saito, H., Yoshizawa, T., 2015. Nanodiamond Finding in the Hyblean Shallow Mantle Xenoliths. *Scientific Reports* 5, 10765. 10.1038/srep10765.
- Tonarini, S., Armienti, P., D'Orazio, M., Innocenti, F., 2001. Subduction-like fluids in the genesis of Mt. Etna magmas: evidence from boron isotopes and fluid mobile elements. *Earth and Planetary Science Letters* 192, 471-483. 10.1016/s0012-821x(01)00487-3.
- Tonarini, S., Agostini, S., Doglioni, C., Innocenti, F., Manetti, P., 2007. Evidence for serpentinite fluid in convergent margin systems: The example of El Salvador (Central America) arc lavas. *Geochemistry, Geophysics, Geosystems* 8, n/a-n/a. 10.1029/2006GC001508.

Reviewer #2 (Remarks to the Author):

In this manuscript the authors attempts to demonstrate, based on new geophysical data that serpentinite diapir coming from the lower plate exhumed through the upper plate and potentially participate to the active Etna volcanism.

The geophysical description of the serpentinite diapir is quite convincing and this is clearly an important discovery for the understanding of the mediterranean dynamics and related volcanism. I'm pretty sure that this data merits to be published in an interntaional journal suchas Nature

However, in its present form the manuscript suffers of weaknesses and merit to be modified before acceptance.

The first point concerns the title : it is too general and not enough informative: key word such as

serpentinite diapir , Calabrian arc are missing

the end of the abstract and the end of the discussion too is too general : you never discuss the seismicity nor the link with continental collision. This is out of the topic

The nature of the lower plate : oceanic, Oceanic continent Transition (OCT) has to be clearly discussed in the light of your data but also in the light of available data are to be clearly discussed. Moreover, in the literature the exact terminology is OCT and not COT

One of the weakness but interesting point of this manuscript concerns the potential link between Etna volcanism and dehydration of serpentinites diapir : to discuss this point you have to show an enlarge schematic cross section showing the deep slab below the Etna volcanoes. Moreover, the geochemistry of the Etna magma especially in terms of volatile elements has to be discussed in the light of the volatile compositions of serpentinites available in the literature (e.g. Deschamps et al., 2013 you cited, but also Savoy, Hattori, Scambelluri...)

in detail

line 29 : If subduction is close to collision : what is the meaning of this sentence ??

line 54 : Tethyan

line 101 : precise the density of the serpentinite diapir : 2.7 ? 2.9 ?? and consequently the percentage of serpentinisation

If the estimated density is higher than the surrounding rocks, you have to discuss how the diapir moves upward ..

line 158 : add reference

line 175-18 : OCT rather than COT

line 186-198 : rephrase all this paragraph describing precisely the geochemistry and add a general cross section

line 208-211 : out of the topic

Figures are ok

REVISION NOTES

We thank the Editor and Reviewers for their very detailed and careful comments that significantly improved our paper. We have taken into consideration all the reviewers' suggestions and modified text and figures accordingly.

Before describing in detail, point by point, how we modified the manuscript and addressed reviewer's comments, we would like to list here main changes and new data we added to the manuscript.

- 1) We have added available Vp/Vs data and plotted the available Vp/Vs profile on our seismic sections in support of the presence of serpentinite in the African plate upper mantle. Vp/Vs data suggest the presence of a 8 km thick serpentinite layer in the CA foreland; these serpentinites are located at a depth fitting well with the diapirs source region.
- 2) We have added new data on grav/mag modelling. We verify our interpretation about the serpentinite diapir providing additional forward models based on different nature of the intruding body: i) mud diapir; ii) Gabbroid cumulates iii) classic oceanic sequence formed by mantle peridotites, gabbroic cumulates and basaltic dikes. These additional forward models have been built considering susceptibility and density values coming from pioneering studies based on direct drillings of lower crust (i.e Leg 176-hole 735B). In all cases, the results of the modeling provided computed magnetic and gravity patterns completely or in part not matching the measured data. These results have permitted to reject different interpretations supporting our prior interpretation of serpentine intrusion below the Ionian Sea.
- 3) We have added available heat flow data in the Ionian Sea. These data suggest that heat flow is low in the study region implying that no thermal anomaly is present in the Ionian basin.
- 4) We have added pore water data from 3 new cores collected above two different serpentinite diapirs and a reference sediment core collected in a sedimentary basin not affected by diapirism. This reference core is used to compare "normal sedimentation" to sediments and fluid flow above the diapirs. The observed pore water signatures indicate active fluid migration related to subsurface dewatering and associated processes. Similar geochemical anomalies have also been found in fluids emanating at some serpentinite mud volcanoes of the Mariana subduction zone, where slab-derived fluids drive serpentinitization and mud volcanism (e.g. Mottl et al., 2004; Hulme et al., 2010).
- 5) We have added information from the literature on xenoliths from the Hyblean plateau. They show evidence of abyssal-type hydrothermal metasomatism, including serpentinitization of ultramafic rocks and represent a direct evidence of the Hyblean plateau being a relic of an ultraslow-spreading Mesozoic Ionian-Tethys Ocean (Scribano et al., 2006; Manuella et al., 2013; Simakov et al., 2015). Some Iblean xenoliths consist of metasomatized gabbroic rocks of lower crustal origin probably derived from an oceanic or ocean-continent transitional environments (Scribano et al., 2006). Hydrothermal modification of xenoliths also suggests that a serpentinite-hosted hydrothermal system (Scribano et al., 2006) typical of slow and ultra-slow spreading ridges was active since the Middle Triassic, as deduced by U/Pb analyses of hydrothermal zircons (Sapienza et al., 2007). These results, backed by geophysical data suggest an oceanic-type lithosphere beneath the Iblean Palteau and may imply some connection between sub Ionian Sea and sub Iblean lithosphere. As suggested by reviewer 1, this evidence, combined with geophysical imaging, is perhaps the stronger evidence of diapirs being made of serpentinite inherited from the lower plate.

- 6) We have added information from the literature on serpentinite diapirs in Eastern Sicily, close to the study region.

If only gravity and magnetic data are available it is not trivial to distinguish between serpentinite and magmatic cumulates especially if the geometry of such cumulates is not known. However, in this work, we know the geometry of the diapirs since deep penetration and high quality depth seismic section are available as stated by reviewer 1. The presence of a thick pile of sediments allow a very good penetration enhancing the recognition of the real geometries of such intrusions. Moreover, in our study, we can use and combine a multidisciplinary dataset including gravity, magnetics, refraction and Vp/Vs data, heat flow, onland geology, occurrence of serpentinite diapirs in the surrounding areas, pore water data, sediment samples and Iblean plateau xenoliths. Combining all these different and independent information, we can suggest that the most plausible composition of the diapirs is serpentinite.

I list here the main evidences on serpentinite diapirs that are discussed in these notes:

- 1) Gravity cores: new data included in this version of the manuscript show an anomalous sediment facies and pore water signature above the diapirs (POINT 2) which stress the occurrence of freshening and fluid flow from deep levels. The comparison of sediment facies and PW above the diapirs and those from a reference core in a region not affected by diapirism favors these conclusions (POINT 2 this file, new Figure 7, lines 118-143, SM6, SM7).
- 2) Gravity anomalies (POINT 3 this file)
- 3) Magnetic anomalies (POINT 3 this file)
- 4) Multichannel seismic images pointing for diapiric structures sourced by deep levels below the crust as stated by the reviewer (Figures 3, 6, 8 and discussion in the manuscript, lines 116-118).
- 5) High Vp/Vs ratio (POINT 5 this file, Figures 6 and 8, lines 102-117 in the manuscript)
- 6) Heat flow (no volcanic intrusions – low heat flow) (POINT 6 this file and Fig. 2 in the manuscript)
- 7) Seismicity (POINT 7 this file)
- 8) Mt. Etna xenoliths and occurrence of serpentinite diapirs in eastern Sicily (POINT 8 this file, lines 197-219 in the manuscript)

Response (*in blue*) to comments by Editor and Reviewers (*in black*).

Ref.: manuscript: #NGS-2017-02-00305 from Nature Geoscience to Nature Communications
Title: Lower plate mantle diapirism in a convergent boundary
Authors: Polonia A., Torelli L., Gasperini L., Cocchi L., Muccini F., Bonatti E., Hensen C., Schmidt M., Romano S., Artoni A., Carlini M.

COMMENTS FROM REVIEWERS

Reviewer #1 (Remarks to the Author):

Parts modified or added to address Reviewer #1 suggestions are marked in yellow.

POINT 1

This paper presents new multibeam, and multichannel seismic reflection, magnetic and gravity field data and core fluid geochemistry to support the presence of mantle-derived serpentinites deriving from the lower plate in the external subduction system of the Calabrian Arc. If it had been more convincingly demonstrated the presence of mantle-derived serpentinite related to subduction processes, this would be, to my best knowledge, one of the first report of the presence of subduction-derived serpentinite in the lower plate of a subduction system. Although the existence of serpentinites in the lower plate it is an interesting discovery, it is unclear what the impact of this finding would be in the wider field of the dynamics of subduction zones, or, alternatively, in our understanding of the mechanisms of mantle exhumation in OCT zones.

The newly discovered serpentinite diapirs within the Calabrian Arc accretionary wedge are the first example of inherited serpentinites formed during Mesozoic rifting, carried below the subduction system by plate convergence processes and raised at shallow levels through lithospheric faults driving margin segmentation and plate boundary re-organization. For this reason, despite the serpentinites are located within the accretionary complex, their formation is not directly linked to subduction processes as for many other convergent setting serpentinites. Plate convergence is the mechanism through which African plate serpentinites have been transferred to the active plate boundary, while transverse trans-tensional lithospheric faults segmenting the Calabrian Arc act as windows along which inherited serpentinites rise in the sub-seafloor.

We have tried to better underline this important concept, which was not properly addressed in the original version of the manuscript. Main changes were included in the abstract and discussion (lines 20-24, 169-171, 181, 284-286).

The main strength of the paper is the unique multibeam, seismic reflection dataset of the Calabrian Arc, which provides very detailed imaging of the structure of this complex Mediterranean subduction system.

Thanks for this comment. A multiscale and multidisciplinary approach aimed at collecting as much evidence as possible to unravel the interplay between geodynamics, tectonics and sediment remobilization along the complex Africa/Eurasia plate boundary in the study region.

POINT 2

The main weaknesses of the paper are three:

(i) Evidence of lower plate mantle diapirism: The paper present not evidence strongly supporting the existence of serpentinite diapirs, although I agree that geophysical data may well be interpreted as serpentinite diapirs. Unfortunately, as concluded by the authors, the fluid geochemistry evidence presented in the paper is not conclusive of active or past mantle serpentinization.

Even though pore water data do not provide unequivocal evidence of active fluid flow from a mantle source, a slight decrease in chlorinity, typical of fresh water mobilization, and release of fluid mobile elements such as Li, B, and Sr (e.g. from clay-mineral transformation) suggests a slight deviation from normal diagenetic trends indicative of dewatering processes underneath. The observed pore water signature was not clear enough to identify a specific process, but it generally supports fluid mobilization at depth.

New cores acquired in the study area were studied and included in this version of the manuscript:

- 1) We have considered a new reference core collected in a sedimentary basin not affected by diapirism and characterized by normal sedimentation. We have made a comparison between this reference core and that collected ontop the diapir (SM7).
- 2) Pore water geochemistry of two further cores collected above the diapirs were analysed and included in the manuscript (lines 118-143).

Location of new cores included in the revised manuscript

COMPARISON BETWEEN REFERENCE CORE (CQ14-02) AND CORE ABOVE THE DIAPIR (GH15-03)

a) CORE CQ14-02

b) CORE GH15-03

Core GH15-03. Left side: core photograph and overprinted magnetic susceptibility in yellow while white boxes represent PW samples described below. Right side: zooms of core sections shown on the left. The recentmost sediments (zooms a, b and c) appear less deformed and represented by parallel laminated units with different colours. Moving downwards in the core (zooms c, d and f) sediment disruption increases and this is accompanied by small scale evidences of fluid flow such as a patchy cloudy facies (Panieri et al., 2013), discontinuities in sediment layers, layers disruption, mud injections and vertical fluid paths which produce sediment homogenization as evidenced by magnetic susceptibility in the bottom part of the core.

- a) the reference core (on the left side) is characterized by an alternation of pelagic sediments and turbidite beds. Sediment layers appear to be undisturbed and continuous. The sharp sandy basal units of the turbidite beds are continuous (left side: zooms 2, 3, 4) and the uppermost muddy units of the turbidites show evidences of undisturbed planar millimetric laminations not affected by sediment disruption (Figure above, zoom 1).
- b) the core collected in the diapiric field area 2 (right side) show clear evidence of sediment layering disruption increasing towards the base of the core. The recentmost sediments (zooms a, b and c) appear less deformed and represented by parallel laminated units with different colours. Moving downwards in the core (zooms c, d and f) sediment disruption increases and this is accompanied by small scale evidences of fluid flow such as a patchy cloudy facies, discontinuities in sediment layers, layers disruption, mud injections and vertical fluid paths which produce sediment homogenization as evidenced by magnetic susceptibility in the bottom part of the core.

NEW PORE WATER DATA:

Comparison between the reference core CQ14-02 (black dots) and cores collected above the diapirs (blue and red dots).

The observed pore water signatures from the two cores above the diapirs indicate active fluid migration related to subsurface dewatering and associated processes. For detailed description see lines 118-143 in the manuscript.

POINT 3

Lacking such evidence, the authors claim the existence of mantle-derived serpentinite based on gravity and magnetic modeling; however, this modelling is very dependent on the forward modelling approach and parameters, and usually does not provide firm evidence to discriminate mantle-derived serpentinite from other lithologies (e.g., Escartin & Cannat, 1999; Dunn et al., 2007; and references therein).

We agree with the reviewer that, usually, forward models are dependent on the approach and on chosen parameters. However, our multidisciplinary study aimed at using independent information from different datasets to better constraint modelling parameters. Geometry of the different modelled bodies and their densities, in fact, were derived from multichannel seismic data, thus reducing uncertainties intimately related to the forward modelling procedure.

Here we describe in more detail our approach and parameters derived from independent data (see also SM5). Interpretation of potential field data can follow several approaches:

- i) in FFT domain trying to identify the depth positioning of causative sources (i.e Euler deconvolution and tilt derivative);
- ii) providing quantitative model inverting the signals using 3D algorithm;
- iii) modelling the recorded signal starting from a geologic baseline (i.e depth migrated seismic section) following a forward approach.

In our study, considering the inhomogeneous distribution of data and taking into consideration the possibility of combining potential field and seismic data, we applied an interpretative technique based on a joint grav-mag forward model. A forward model allows computing synthetic magnetic or gravity profiles generated by a set of different causative bodies acting on their physical parameter as density, susceptibility or both. This kind of modelling is thus based on 2 free parameters:

- 1) geometry of causative bodies;
- 2) their physical properties.

Without any additional constrain, results of forward modelling are dependent on the choices made by the user providing not unique solutions.

In this study we strengthen our interpretation fixing different constrains in the modelling approach:

1) Our model is based on the geological evidence derived from the depth migrated seismic sections (line CA-A). The geometry (shape and depth position) of the seismo-stratigraphic units reported in Figure 4 (manuscript) were derived directly from the interpretation of the seismic section. We fixed geometry of the seismo-stratigraphic units (including the intrusive body) and then modelled the gravity and magnetic patterns changing only density and susceptibility values.

2) We performed a 2.75 D gravity and magnetic model using GmSYS OASIS geosoft software.

2.75D means that we have the possibility to model and constrain geometry and physical parameters of bodies across the profile improving the best fitting process. This approach is preferred (and often required) in the case of peculiar geological settings characterized by high frequency signals and lateral variability of lithology such as our study area. On the contrary, the 2D model supposes an infinite length across the profile for any modelled body. This kind of approach would be preferable for modelling large-scale geologic features, such as an oceanic ridge.

3) The density values used in the forward model derived from the inversion of seismic velocities using Gardner's equation.

4) Magnetic profiles were modelled starting from the average susceptibility values of main lithologies forming the mid-shallow Ionian crust derived from the interpretation of seismic data and refraction profiles (De Voogd et al., 1992).

5) The forward model has been further constrained by the simultaneous fitting of gravity and magnetic profiles. A joint magnetic and gravity interpretation increases the reliability of the modelling permitting to reduce the number of potential solutions.

Concluding, in our forward modelling approach a multidisciplinary dataset providing independent information, reduced uncertainties with final solutions that are better constrained relative to other studies where different datasets are not available.

POINT 4

... This is well known in slow spreading mid-ocean ridges where, to date, gravity and magnetic modeling has been unable to provide solid evidence to unambiguously discriminate serpentinization at depth from other combinations of mafic and ultramafic rocks (e.g., Escartin & Cannat, 1999; Dunn et al., 2007; and references therein). Gravity and magnetic modeling can also be consistent with magmatic cumulates, making it difficult to distinguish between gabbro and serpentinized peridotite (e.g., Escartin & Cannat, 1999; Dunn et al., 2007; and references therein).

The main challenge of forward modelling was focused on the interpretation of the anomaly generated by the intrusive body. Simultaneous best fitting of grav-mag data addressed a peculiar value of density (2.7 g/cm³) and low susceptibility (0.015) for this intrusive body. Such values point to the diapir being made of serpentinites (see also SM5).

Here, we stress our final interpretation, providing more details on the different models carried out to understand its lithology and origin: a) mud diapir, b) cumulates of mantle peridotites, c) gabbroid intrusive, d) swarm of basaltic dykes.

a) Mud Diapir:

We model the intrusive body as due to mud diapirism (i.e. Camerlenghi et al., 1995). Modelling was performed using a density value = 2.2 g/cm³ (density value of quaternary sediments) and susceptibility = 0.

Recovered gravity profile does not match the observed signal with a maximum misfit error of about 47.1 mGal. (Fig R1). If we assume a lower density of 1.75 to 1.92 g/cm³ as provided by a set of laboratory measurements of bulk density for mud volcanoes in the Mediterranean ridge (see Tab.4 in Camerlenghi et al., 1995) the misfit error of the modelled gravity anomaly increases further.

Figure R1 - Results of gravity (left) and magnetic (right) forward model considering a mud diapir

Magnetic modelling provides a pattern different from the observed anomaly with a misfit value of 36.2 nT (Fig R2). Both gravity and magnetic observations allow us to reject this hypothesis.

b) Gabbroid cumulates hypothesis:

We take into account the possibility of a gabbroic nature for the intrusive body, since the value of the magnetic susceptibility of gabbros and serpentine has wide areas of overlapping. The serpentinization of ultramafic rocks overturns radically the magnetic properties of the host rock producing serpentinized rocks with susceptibility ranging from 10^{-4} to 10^{-1} SI (Oufi et al., 2002; Maffione et al., 2014). In the case of gabbroic rocks the susceptibility also shows high variability (Figure R3), ranging from low values (i.e $1.72 \cdot 10^{-2}$ SI in Richter et al, 1996; Dick et al., 1999) up to values comparable to those of basalts (Gee and Kent, 2007 and references therein). Gee and Kent (2007) also stated that “Early estimates of the gabbroic contribution to marine magnetic anomalies relied on dredged samples and studies of ophiolites, with the results from ophiolites yielding variable and sometimes contradictory results”.

Figure R1 - Susceptibility vs depth plot, of gabbroic sequence - ODP leg176, Hole 735 B (modified from Dick, et al., 1999), data coming from “http://www-odp.tamu.edu/publications/176_IR/VOLUME/TABLES/CHAP_03/03_14.TXT”

Magnetic forward modelling obtained considering the intrusive structure made of gabbros is shown below (Fig. R4). We used the susceptibility values of gabbroid sequences from Dick et al. (1999) and Richter et al. (1996). Two computed profiles differ from the observed signal but, the average misfit errors (-29/12 nT for 0.007 SI and

-8/+15 nT for 0.017 SI) are similar to those obtained for the hypothesis of serpentinites (see below). This confirms that, gabbros and serpentinites could not be told apart only by using the magnetic data, in agreement with the reviewer observation.

Figure R2 - Results of magnetic forward modelling obtained considering the intrusive body made in gabbro.

On the other hand, the density of gabbros is well constrained in the range of 2.9 g/cm³ (Hyndman and Drury, 1976; Manea and Manea, 2008) up to 2.95 g/cm³ (Carlson and Raskin, 1984). Results of drilling a gabbroid sequence from site 785 of ODP leg 176 (Dick et al., 1999) confirmed again a very stable average density of 2.98 g/cm³ (see fig. R5 redrawn using data from Leg 176-hole 735B) for mafic gabbroid rocks.

Figure R3 - density vs depth from drill site 735B - Leg 176 (Dick et al., 1999). Data from http://www-odp.tamu.edu/publications/176_IR/VOLUME/TABLES/CHAP_03/03_15.TXT

Such high density values for the intrusive body in our modelling results in a gravity anomaly not fitting the acquired data, producing a maximum discrepancy larger than 15 mGal (fig. R6).

Concluding, the combined analyses of the two independent datasets (gravity and magnetics) is very useful to discriminate the nature of the intruding body suggesting that the hypothesis of an intrusive gabbroid nature is unlikely for the diapir.

Figure R4 Results of forward gravity model obtained considering the intrusive body made in gabbros.

c) Basaltic-like dykes swarm hypothesis

We then modelled the intrusive body as related to a classic sequence of oceanic crust in a typical low spreading center (Gee and Kent 2007, Dunn and Forsyth, 2007) where mantle peridotites are covered by gabbros, dykes swarm and lava flows. This hypothesis is not well constrained in the study area because there is no evidence of shallow eruptive manifestation or emplacement of lavas. Since we cannot discard *a priori* this hypothesis, we carried out a forward magnetic and gravity modelling considering a sequence of ultramafic and mafic rocks as reported in figure R7.

We assume low serpentinized mantle peridotites at the base of an intrusion characterized by low susceptibility (0.001 SI) and high density (3.0 g/cm³) (Maffione et al., 2014, Hyndman and Drury, 1999, Fichler et al., 2011). We accounted for the gabbroid sequence using average susceptibility of 0.0017 SI from Richter et al. (1996) and density 2.95 g/cm³ from Hyndman and Drury (1999). The basaltic dikes sequence has been modelled using a density of 2.85 g/cm³ (Staudigel, 1994 and reference therein) and high susceptibility (average 0.05 SI) (Larson et al., 1992, Tanaki et al., 1992; Varga et al., 2004). Results are reported in the following figures.

Both magnetic and gravity calculated profiles (fig. R7 left and right upper panels, respectively) differ from those observed with a maximum misfit error of 9.8 mgal and 98 nT, respectively. The hypothesis of an intrusive ensemble made of gabbro and basaltic dikes seems unlikely.

Figure R5 - Lower panel: sketch of oceanic sequence used to model the intrusive body. Upper left panel: results of forward magnetic model; upper right panel: results of gravity modeling.

d) Serpentinites hypothesis

Finally, the values used in the joint modelling of gravity and magnetic data are compatible with the presence of a serpentinized peridotite intrusive body. We have included the seismic section below the modelling parameters.

Degree of serpentinization:

In the revised version of the manuscript, we have added in the Supplementary files a further consideration based on two different assumptions:

- a) the serpentinization of peridotites which are formed by paramagnetic minerals (Olivine+pyroxene) produce of an increase in susceptibility because magnetite forms during serpentinization by the breakdown of the iron-rich olivines (fayalites). Susceptibility of serpentinites is also dependent on the amount of fayalites in the original peridotite (Bronner et al., 2011). The relationship between increasing magnetization vs degree of serpentinization is not linear (Toft et al., 1999, Maffione et al., 2014 and reference therein).
- b) the serpentinization process is responsible for a decrease in density, which follows a linear trend as function of the degree of alteration (Toft et al., 1990).

Considering that serpentinization acts on the physical parameter of the original peridotites we can use our forward model to deduce serpentinization degree. Our model suggests that the intruded body has density in the order of 2.7 g/cm^3 . Following the equation proposed by Miller and Christens (1997), such value is indicative of a serpentinization of about 75%.

This is confirmed by an independent observation based on magnetic data. Maffione et al. (2014) proposed a general relationship between susceptibility and serpentinization degree. For susceptibility of 0.015 SI as deduced in our model, the resultant serpentinization degree is 68%.

These values, obtained again with two independent datasets, are similar to those obtained from the Vp/Vs data (70% for S1 layer) as reported by D'Alessandro et al (2016). This agreement in the percentage of serpentinization confirms the hypothesis of intrusion of serpentines material below the Ionian seafloor as suggested by Vp/Vs data (see also POINT 5 below).

POINT 5

Combined P- and S-wave velocity profiles would have been necessary to assess if the seismic structure is compatible with variable abundance of mafic and mantle-derived serpentinite (e.g., Carlson and Miller, 1997).

We fully agree with the reviewer. We have added some data and more discussion on this critical point, which was possibly not fully addressed in the original version of the manuscript (lines 102-118, Figures 5 and 8).

P- and S- wave profiles are available in the southern region of the diapiric field (D'Alessandro et al., 2016). Simultaneous inversion of P- and S-wave arrival times, collected during a 3-years OBS/H monitoring campaign, yields 1D P- and S-wave velocity model for the Ionian lithosphere (the location of the three OBS-H instruments has been added in Fig. 1 (green stars) and reported below). The 1-D model highlights the presence in the Ionian upper mantle of two layers characterized by high seismic P-wave velocity, as shown in the figure below modified from D'Alessandro et al. (2016).

These two layers, with thicknesses of about 3.3 Km and 5 km, respectively, and ranging from ~8 to ~16 Km in depth, are characterized by low S-wave Velocity (S1 = 3.05–3.2 km/s, S2 = 3.85 km/s) and high values of Vp /Vs (S1=2.06–2.09, S2= 1.95).

- S1: Vp=6.3–6.7 Km/sec, Vs=3.05–3.2 km/s, Vp /Vs = 2.06–2.09
- S2: Vp=7.5 Km/sec, Vs= 3.85 km/s, Vp /Vs = 1.95

These two layers are interpreted as partly serpentinized peridotite (S1: 55-65% serpentinization of the upper mantle while S2 is consistent with 15-25% serpentinization).

These values of V_p/V_s for serpentinites are in agreement with other data in the literature including references suggested by the reviewer (i.e. Carlson and Miller, 1997).

[Redacted]

The figure on the right represent the location of OBH-S instruments (green stars) used during the 3-years monitoring campaign.

Comparing these results with our pre-stack depth migrated multichannel seismic data acquired in the same region (below), we find that there is a good correlation between V_p/V_s data derived serpentinite layers and our interpretation of the Ionian lithosphere. In particular, the two serpentinized layers envisioned from the V_p/V_s data fit well the source area of the diapirs.

I agree, though, that the geometry and apparent rooting in the Moho, supports serpentinization, but does not exclude they can be mafic to ultramafic magmatism intrusions.

We think that the geometry and rooting in the Moho of the diapirs combined with the presence of two layers with high Vp/Vs ratio at the same source depth supports a serpentinite origin for the rising material. We have added the Vp/Vs profile in the study region on our multichannel seismic line and we thank the reviewer for this useful suggestion, which strengthens our interpretation.

POINT 6 –

Heat flow data show that the thermal regime of the Ionian basin is lower than normal (Della Vedova and Pellis, 1986). In this paper eleven heat flow measurements, with in situ determination of thermal conductivity were presented (table below). In the deepest region of the Ionian basin the mean heat flow value is 31.8 +/- 5.0 (S.D.) mW m⁻², a value close to the 33.5 mW m⁻² measured at DSDP 374. Measurements in a region close to the diapir field area is even lower (27.0 +/- 9.1 mW m⁻²). This value does not support magmatic intrusions in the study region.

We have added a comment on these data in the manuscript (lines 89-90) and also in **Figure 2** (see below).

TABLE I. Summary of heat flow measurements in the Ionian Sea (Bannock 9/84 cruise).

STATION	LAT. N	LONG E	DEPTH (m)	B.W.T. (°C)	PEN (m)	PR. (N)	GRADIENT (mK m ⁻¹)	CONDUCTIVITY N (W m ⁻¹ K ⁻¹)	H.F. (mW m ⁻²)		
GT 84-1	36° 45.0'	16° 21.2'	3280	14.0	7.0	4	26.1±5.2	--	1.1*	29	SW Messina Cone
GT 84-2	36° 40.7'	15° 56.6'	3285	13.9	7.0	4	30.3±5.0	2	1.17	35	
GT 84-3	36° 33.7'	16° 33.0'	3400	14.0	6.5	4	16.2±2.7	2	1.09	18	
GT 84-4	36° 26.2'	15° 56.6'	3345	14.0	6.5	4	34.4±4.0	--	1.05*	36	
GT 84-5	36° 01.1'	15° 57.9'	3685	14.0	6.5	4	15.7±1.9	--	1.1*	17	
GT 84-6	35° 39.1'	16° 45.0'	3925	14.1	4.5	3	29.5±4.4	2	1.04	31	Ionian abyssal plain
GT 84-7	35° 31.8'	16° 57.0'	3980	14.1	6.5	4	31.6±0.7	3	1.24	39	
GT 84-8	35° 28.1'	17° 18.5'	3950	14.1	6.5	4	29.8±5.3	2	1.16	35	
GT 84-9	35° 45.6'	18° 21.3'	4085	14.2	6.5	7	24.4±6.3	3	1.01	25	
GT 84-10	35° 58.5'	18° 19.6'	4085	14.2	7.0	7	27.6±8.5	3	1.03	28	
GT 84-11 I	36° 13.0'	18° 25.2'	4070	14.2	7.0	5	30.2±7.0	--	1.1*	33	
II	36° 13.3'	18° 23.5'	4070	14.2	7.0	5	28.6±4.6	2	1.13	32	

Data from della Vedova and Pellis, 1986 (CIESM congress, Palma di Majorca)

- 35: 30.3+/-5.0 mW m⁻²
- 36: 34.4+/-4.0 mW m⁻²
- 17: 15.7+/-1.9 mW m⁻²
- 29: 26.1+/-5.2 mW m⁻²
- 18: 16.2+/-2.7 mW m⁻²

New figure 2 in the manuscript with location and reported heat flow measurements in the study area.

POINT 7 - SEISMICITY

[Redacted]

Seismicity in the western Ionian Sea is diffuse with a major epicentre density in the region of the Alfeo-Etna fault system and serpentinite diapirs (SgROI et al., 2007; 2014; Polonia et al., 2016).

SgROI et al., (2007; 2014) have analyzed 239 high-quality events and the waveform features and the possible source zones for those events are investigated by means of polarization and particle motion techniques. Most of the 239 events (213) are characterized by high values of rectilinearity typical of P- and S-arrival particle motions, while the remaining 26 events have different polarization features, with an emergent first phase and prevalently planar polarization. SgROI et al. (2007) have interpreted the latter signals as being associated with submarine landslides since slope instability signals have high frequency content, lack easily recognizable P- and S-phases,

and have longer durations than earthquake signals. A rose diagram of the back-azimuth values of these events, centered at NEMO-SN1 (black dot in the figure on the left), can help identify the areas of major instability. One of the high frequency signals area, is located exactly in the diapirs field region (SE of the S1 station, see figure on the right); we suggest that this might be an indication of diapir activity enhancing slope instabilities.

(ii) The second weakness of the paper is that the age putative serpentinite diapirs, their genesis, and their relation to Calabrian subduction zone is not well constrained and highly speculative. As pointed out by the authors, the shallow depth and their plate location make it unlikely that such putative serpentinite diapirs are related to Calabrian subduction processes. They are most likely inherited from Tethyan ocean formation (i.e., Mesozoic in age), as the authors conclude in the highly speculative section “Origin of serpentinites and the Mesozoic Tethyan basin”. If they are inherited Mesozoic features, the title and the main contribution of the paper are rather misleading because there is not relationship between “Lower Plate Mantle Diapirism” and “Convergent margin” processes, a relationship one might expect to be the main contribution of this report from its title.

The Calabrian Arc is a convergent margin and serpentinite diapirs are located within the inner accretionary wedge of the subduction complex. This does not mean that they are directly related to subduction processes and, in fact, in our reconstruction, they are inherited serpentinites in the lower plate rising thanks to trench-orthogonal rifting processes.

The transtensional faults along which the putative serpentinites rise to shallow levels, are related to slab rollback and tearing (Polonia et al., 2016). These faults are segmenting the continental margin and accommodating the re-organization of the plate boundary, which causes incipient rifting processes in the accretionary wedge.

Summarizing: the newly discovered diapirs within the Calabrian Arc accretionary wedge are made of inherited serpentinites formed during Mesozoic as suggested by Vp/Vs data, carried below the subduction system by plate convergence processes and raised at shallow levels within the accretionary wedge through lithospheric faults driving margin segmentation and plate boundary re-organization driven by slab rollback. For this reason, despite the serpentinites are located within the accretionary complex, their formation is not directly related to subduction as in many other convergent setting serpentinites. Plate convergence is the mechanism through which serpentinites have been transferred to the active plate boundary. Trans-tensional lithospheric faults segmenting the Calabrian Arc act as windows along which inherited serpentinite rise in the sub-seafloor. We have underlined better this point (this part has not been added to the revised manuscript, but we can add it if useful).

Most of this section speculates about the formation of such diapirs during Mesozoic rifting of the Tethys Ocean or in OCT in a non-convergent setting. The authors should refer to it as “Inherited, Lower Plate Serpentinite Diapirism” unrelated to the Convergent margin processes. On the other hand, the term “Mantle Diapirism” is unfortunate: at best, it is Serpentinite Diapirism. Mantle Diapirism refers to ductile upwelling — usually associated to decompression melting— of mantle-derived peridotite. Serpentinite diapirs are produced by buoyant upwelling or tectonic exposure of serpentinites formed by hydrothermal alteration of non-mantle- or mantle-derived rocks.

Thanks for this comment. We have modified the title accordingly.

The exact mechanisms and age of formation of such diapirs, and their relevance for Calabrian subduction system and for subduction systems overall are not compellingly demonstrated in the manuscript. Throughout the manuscript, the authors overstress the importance of the serpentinites, without really going into the details of the process.

This comment implies that this point was not properly discussed and thus we have added a section on age and mechanisms of serpentinite diapirism (lines 237-247) avoiding to overstress the importance of serpentinites as indicators of the nature of the Mesozoic ocean which, we agree, is highly speculative.

To demonstrate compellingly mechanism and age of formation of the diapirs we would need direct information on the diapirs and hosting sediments. Without samples, unfortunately, the only way to progress is consider indirect evidence based mainly on seismo-stratigraphic analyses of our seismic lines. Age and geometry of the faults driving diapirism was reconstructed through seismo-stratigraphic analyses in Polonia et al. (2016 - Tectonophysics). We have added some data in this new version of the manuscript (lines 237-247).

For instance:

(lines 166-171) “The newly discovered serpentinite diapirs field below the Ionian Sea have important implications. First, diapirs may carry to the surface information about the nature of the crust presently still controversial, and about the geometry and processes that have driven Tethyan ocean rifting”.

Even if they carry such information, how this information is going to be retrieved to understand the nature of the crust? How the geometry and processes that have driven Tethyan ocean rifting are going to be studied? A non-lower plate exposure in a divergent setting (OCT, or back-arc basin) is a much better location to investigate such processes because they will not be masked by the upper plate. Better constrains can be obtained, for instance, from the nearby Tyrrhenian basin, which exposes mantle peridotites and it is undisturbed by subduction (Bonatti, 1999).

We agree with the reviewer that serpentinites in the Tyrrhenian Sea might provide important constraint on spreading and mantle exhumation processes in the back arc region. But this is something not related to our study and beyond the scope of our manuscript. We have described better geodynamic implication of Ionian Sea serpentinites in the new version of the manuscript following also suggestions from reviewer 2 (lines 274-283).

(lines 199-201) “Serpentinite diapirs in the Ionian Sea provide evidence on how mantle rocks may be transferred from rifting, to basin floor to subduction where they can give rise to ophiolites, generally regarded as fragments of subducted oceanic crust emplaced onto continental crust through underplating and tectonic erosion”.

It is uncertain how the evidence provided in the manuscript from the Calabria subduction system can help to understand the origin of supra-subduction ophiolites, which usually provide a magmatic record of supra-subduction magmatism, and discriminate from those formed in mid-ocean ridges.

We removed this part, which was not clear.

(lines 202-204) “Our Ionian Sea findings show that serpentinites may be already in existence before entering the subduction channel and can reach shallow levels of the subduction complex through large-scale diapirism, enhancing shear processes and margin disruption during the final closure of the ocean.”

The authors has not proven that such diapirism is related to subduction processes; actually fore arc serpentinite diapirism is related to active serpentinization, which is not demonstrated here. This conclusion somewhat contradicts their earlier conclusion on the formation of serpentinite diapirs during Mesozoic rifting in a non-convergent setting.

Fore arc serpentinite are not related to subduction nor to active serpentinization as stated in the original manuscript. They are Mesozoic inherited serpentinites in the Ionian deep basin not related to subduction processes. We do not understand why the reviewer is writing: “The authors has not proven that such diapirism is related to subduction processes”. We have never proposed that diapirism is related to subduction processes.

Vp/Vs data show that serpentinites are widespread in the fore arc region and already in existence before entering the subduction system implying that they are inherited from the lower plate. Inherited serpentinite give rise to intrusions as a response of active trench-orthogonal rifting and margin disruption, which is driven by slab roll-back and plate boundary re-organization in the central Mediterranean Sea.

(iii) The last weakness of the paper is the relationship between the putative serpentine diapirism and recent volcanism at Mt. Etna. This link is highly conjectural, lacks solid geochemical evidence, and the cause-effect between serpentine diapirism and volcanisms it is not demonstrated. The relationships between the reported diapiric alignments towards the Hyblean plateau is something worth to investigate in greater depth, but it is not compelling evidence for the role of serpentine in the genesis of the Mt. Etna volcanism.

Thank-you for this comment. The origin of Mt. Etna is beyond the scope of this manuscript; what we intend to underline is the alignment of the serpentinite diapirs and Mt. Etna along the same fault (i.e. Alfeo-Etna fault). This observation suggests a structural control on both processes (i.e. serpentinite diapirism and Etna volcanism). We did not suggest a cause-effect between serpentine diapirism and volcanism. To avoid this misunderstanding we have modified the text (lines 248-267) following reviewer’s suggestions.

POINT 8

There is direct evidence (non-cited) that the unexposed lithospheric roots of the Hyblean plateau consist of relict of ultraslow-spreading Mesozoic Ionian-Tethys Ocean exists in the xenolith suite of the Hyblean tuff-breccia deposits (e.g., Scribano et al., 2006; Manuella et al., 2013). Most of Hyblean xenoliths contain evidence of abyssal-type hydrothermal metasomatism, including serpentinization of ultramafic rocks (Scribano et al., 2006; Manuella et al., 2013; Simakov et al., 2015). I think that this evidence, combined with the geophysical imaging provide in this manuscript, is perhaps the stronger evidence of these alignments being serpentinite diapirs and their inheritance from the lower plate.

Thank-you very much for this comment. The original version of the manuscript did not include this issue because of text limitations. We agree that this point is very important and needs to be addressed in order to reinforce our interpretation. Since the text has been reduced in other parts we have space to describe in more detail the Hyblean plateau xenoliths and their implication (lines 195-216).

A separate issue is a cause-effect relationship between the presence of such serpentinite diapirs and the origin of the Mt. Etna volcanism. Boron isotope geochemistry provides unequivocal evidence of role of serpentinite-derived fluids in the genesis of volcanism (e.g., Straub and Layne; 2002; Tonarini et al., 2007), but, to my knowledge, boron isotope systematic of Mt. Etna volcanism does not support such derivation (Tonarini et al., 2001). Even if cannot be excluded, there is not geochemical evidence for the role of serpentinite in the genesis of Mt. Etna volcanism.

We fully agree with this comment. We did not suggest any cause-effect relationship between serpentinites and the genesis of Mt. Etna volcanism. There is no isotope geochemical evidence, which suggests serpentinite-derived fluids in the genesis of Etna volcanism. What we suggest is a structural control on the genesis of both serpentinite diapirs in the Ionian Sea and Mt. Etna in NE Sicily since they are aligned along the same lithospheric fault. We have clarified this point (line 235-267).

The authors argue the involvement of serpentinite in the genesis of Etna volcanism, mostly on the basis of a fluid-rich environment and its provenance from the lower plate of this volcanism: (lines 196-198) “When the AEF tectonic behavior shifted from compression to extension, the volcanic products of Mt. Etna reached the surface together with serpentinite diapirs, in agreement with the voluminous melting under Mount Etna having originated from under the African plate.”; however, These arguments that do not necessarily evidence the role of serpentinites in the genesis of Mt. Etna volcanism.

We fully agree with the reviewer and, in fact, we did not suggest any cause-effect relationship between serpentinites and the genesis of Mt. Etna volcanism. See answers above.

REFERENCES:

- Bonatti, E., Seyler, M., Channell, J., Girardeau, J., Mascle, G., 1990. Peridotites drilled from the Tyrrhenian Sea, ODP Leg 107, Peridotites drilled from the Tyrrhenian Sea, ODP Leg 107, pp. 37-47.
- Carlson, R.L., Miller, D.J., 1997. A new assessment of the abundance of serpentinite in the oceanic crust. *Geophysical Research Letters* 24, 457-460. 10.1029/97GL00144.
- Dunn, R.A., Forsyth, D.W., 2007. 1.12 - Crust and Lithospheric Structure – Seismic Structure of Mid-Ocean Ridges A2 - Schubert, Gerald, *Treatise on Geophysics*. Elsevier, Amsterdam, pp. 419-443.
- Escartin, J., Cannat, M., 1999. Ultramafic exposures and the gravity signature of the lithosphere near the Fifteen-Twenty Fracture Zone (Mid-Atlantic Ridge, 14 -16.5 degrees N). *Earth and Planetary Science Letters* 171, 411-424.
- Manuella, F. C., Brancato, A., Carbone, S. & Gresta, S, 2013. A crustal–upper mantle model for southeastern Sicily (Italy) from the integration of petrologic and geophysical data. *J. Geodyn.* 66, 92–102 (2013).
- Straub, S.M., Layne, G.D., 2002. The systematics of boron isotopes in Izu arc front volcanic rocks. *Earth and Planetary Science Letters* 198, 25-39.
- Scribano, V., Ioppolo, S. & Censi, P, 2006. Chlorite/smectite-alkali feldspar metasomatic xenoliths from Hyblean Miocenic diatremes (Sicily, Italy): Evidence for early interaction between hydrothermal brines and ultramafic/mafic rocks at crustal levels. *Ofioliti*. 31, 161–171.
- Simakov, S.K., Kouchi, A., Mel’nik, N.N., Scribano, V., Kimura, Y., Hama, T., Suzuki, N., Saito, H., Yoshizawa, T., 2015. Nanodiamond Finding in the Hyblean Shallow Mantle Xenoliths. *Scientific Reports* 5, 10765.

10.1038/srep10765.

Tonarini, S., Armienti, P., D'Orazio, M., Innocenti, F., 2001. Subduction-like fluids in the genesis of Mt. Etna magmas: evidence from boron isotopes and fluid mobile elements. Earth and Planetary Science Letters 192, 471-483. 10.1016/s0012-821x(01)00487-3.

Tonarini, S., Agostini, S., Doglioni, C., Innocenti, F., Manetti, P., 2007. Evidence for serpentinite fluid in convergent margin systems: The example of El Salvador (Central America) arc lavas. Geochemistry, Geophysics, Geosystems 8, n/a-n/a. 10.1029/2006GC001508.

We have included and discussed all these references.

Reviewer #2 (Remarks to the Author):

Parts modified or added to address Reviewer #2 suggestions are marked in green.

In this manuscript the authors attempts to demonstrate, based on new geophysical data that serpentinite diapir coming from the lower plate exhumed through the upper plate and potentially participate to the active Etna volcanism.

The geophysical description of the serpentinite diapir is quite convincing and this is clearly an important discovery for the understanding of the mediterranean dynamics and related volcanism. I'm pretty sure that this data merits to be published in an interntaional journal suchas Nature

However, in its present form the manuscript suffers of weaknesses and merit to be modified before acceptance.

The first point concerns the title : it is too general and not enough informative: key word such as serpentinite diapir , Calabrian arc are missing

Thanks for this comment. The title has been changed accordingly.

the end of the abstract and the end of the discussion too is too general : you never discuss the sesismicity nor the link with continental collision. This is out of the topic

We agree with the reviewer. We have removed this part from the abstract and discussed it in the manuscript (lines 274-283).

The nature of the lower plate : oceanic, Oceanic continent Transition (OCT) has to be clearly discussed in the light of your data but also in the light of available data are to be clearly discussed. Moreover, in the literature the exact terminology is OCT and not COT

Thanks for this comment. We have changed accordingly and included references on the nature of the crust in the Ionian Sea (lines 35-37).

One of the weakness but interesting point of this manuscript concerns the potential link between Etna volcanism and dehydration of serpentinites diapir : to discuss this point you have to show an enlarged schematic cross section showing the deep slab below the Etna volcanoes. Moreover, the geochemistry of the Etna magma especially in terms of volatile elements has to be discussed in the light of the volatile compositions of serpentinites available in the literature (e.g Deschamps et al., 2013 you cited, but also Savoy, Hattori, Scambelluri...).

We would like to stress again that we did not suggest any link between serpentinites and the genesis of Mt. Etna volcanism because, at the moment, there is no isotope geochemical evidence, which suggests serpentinite-derived fluids in the genesis of Etna volcanism. What we suggested in the original version of our manuscript is the structural control exerted by the Alfeo-Etna fault on the genesis of both serpentinite diapirs in the Ionian Sea and Mt. Etna in NE Sicily. Diapirs and Mt. Etna are, in fact, aligned along the same lithospheric fault. We have clarified this point.

Thanks for the suggestion to include a third cross section across the Mt. Etna volcano, which clarifies relationships between the different geodynamic features in this complex area.

New Figure 8 in the revised manuscript

in detail

line 29 : If subduction is close to collision : what is the meaning of this sentence ??

lien 54 : Tethyan

ok

lien 101 : precise the density of the serpentinite diapir : 2.7 ? 2.9 ?? and consequently the percentage of

serpentinisation

If the estimated density is higher than the surrounding rocks, you have to discuss how the diapir moves upward

..

line 158 : add reference

line 175-18 : OCT rather than COT

ok

line 186-198 : rephrase all this paragraph describing precisely the geochemistry ad add a general cross section

line 208-211 : out of the topic

Figures are ok

REVIEWERS' COMMENTS:

Reviewer #1 (Remarks to the Author):

In my first review of the manuscript, I put forward three main weaknesses of the paper. I think that in the revised version of the paper the authors have addressed convincingly my comments. The revised manuscript now provides solid evidence and arguments for the presence of serpentinite diapirs in the lower plate of a subduction zone. It also makes it clear that these diapirs are inherited from the Mesozoic rifting event. The authors also addressed and I agree on their reply to most of my other minor comments. Below, I just briefly comment on the modifications the authors made to address my three main comments.

1) Geophysical evidence for the presence of serpentinite diapirs. The manuscript now provides new Vp/Vs data and plotted the available Vp/Vs data to support of the presence of serpentinite diapirs. The revised manuscript now provides a more robust forward modelling and a more accurate discussion. New geophysical data and fluid data are very helpful and make more compelling their hypothesis. The only minor comment is that they used a layered lithological model for fast-spreading mid-ocean ridge that is contradictory with the derivation from a slow-spreading mid-ocean ridge. Anyhow, as they argue, their geometry is well constrained by their high-resolution seismic data. The authors has changed the title of the paper, which is now a more in line with the main finding of the paper.

2) The second weakness of the paper was that the age putative serpentinite diapirs, their genesis, and their relation to Calabrian subduction zone was not well constrained and highly speculative. Main issues are now rewritten and clarified. In the revised version, the authors have omitted and rewritten the more speculative paragraphs; the discussion is now smoother and clarify many of these issues.

3) My third main comment was about the relationship between the putative serpentine diapirism and recent volcanism at Mt. Etna. In the revised version the authors has clarified the main issues I raised; they have rewritten this part of the manuscript to avoid any potential misunderstanding about the evidence for presence of serpentinite diapirs and the origin of Mt. Etna volcanism. The revised manuscript no focuses on the sublithospheric structure and the hypothesis that the alignments towards the Hyblean plateau are serpentinite diapirs. As suggested in my previous review, they have added existing evidence for the presence of mantle xenoliths that strongly this evidence. AS I pointed out, combined with the geophysical data, is perhaps the stronger evidence that these alignments are serpentinite diapirs inherited from the lower plate.

The result of the revision is a more compelling contribution that, to my best of my knowledge, constitutes the first evidence for the subduction of serpentinite diapirs in a lower plate formed in a slow-spreading mid-ocean ridge. Certainly, this will become a reference for the future studies of subduction zones and the Mediterranean Sea.

Dr. Carlos J. Garrido

Reviewer #2 (Remarks to the Author):

I review for the second time this interesting and well documented manuscript. I consider that you take into account my remarks but also the remarks of the second reviewer.

I have only two very small suggestions :

line 22 page 1 : add The serpentinites , probably formed

in the figure 8 : add the vertical scale, concerning the serpentinitized layer S1 and S2 they are probably not continuous, as in slow spreading ridges, serpentinites are patchy at the sea floor

REVIEWERS' COMMENTS:

Reviewer #1 (Remarks to the Author):

In my first review of the manuscript, I put forward three main weaknesses of the paper. I think that in the revised version of the paper the authors have addressed convincingly my comments. The revised manuscript now provides solid evidence and arguments for the presence of serpentinite diapirs in the lower plate of a subduction zone. It also makes it clear that these diapirs are inherited from the Mesozoic rifting event. The authors also addressed and I agree on their reply to most of my other minor comments. Below, I just briefly comment on the modifications the authors made to address my three main comments.

1) Geophysical evidence for the presence of serpentinite diapirs. The manuscript now provides new Vp/Vs data and plotted the available Vp/Vs data to support of the presence of serpentinite diapirs. The revised manuscript now provides a more robust forward modelling and a more accurate discussion. New geophysical data and fluid data are very helpful and make more compelling their hypothesis. The only minor comment is that they used a layered lithological model for fast-spreading mid-ocean ridge that is contradictory with the derivation from a slow-spreading mid-ocean ridge. Anyhow, as they argue, their geometry is well constrained by their high-resolution seismic data. The authors has changed the title of the paper, which is now a more in line with the main finding of the paper.

2) The second weakness of the paper was that the age putative serpentinite diapirs, their genesis, and their relation to Calabrian subduction zone was not well constrained and highly speculative. Main issues are now rewritten and clarified. In the revised version, the authors have omitted and rewritten the more speculative paragraphs; the discussion is now smoother and clarify many of these issues.

3) My third main comment was about the relationship between the putative serpentine diapirism and recent volcanism at Mt. Etna. In the revised version the authors has clarified the main issues I raised; they have rewritten this part of the manuscript to avoid any potential misunderstanding about the evidence for presence of serpentinite diapirs and the origin of Mt. Etna volcanism. The revised manuscript no focuses on the sublithospheric structure and the hypothesis that the alignments towards the Hyblean plateau are serpentinite diapirs. As suggested in my previous review, they have added existing evidence for the presence of mantle xenoliths that strongly this evidence. AS I pointed out, combined with the geophysical data, is perhaps the stronger evidence that these alignments are serpentinite diapirs inherited from the lower plate.

The result of the revision is a more compelling contribution that, to my best of my knowledge, constitutes the first evidence for the subduction of serpentinite diapirs in a lower plate formed in a slow-spreading mid-ocean ridge. Certainly, this will become a reference for the future studies of subduction zones and the Mediterranean Sea.

Dr. Carlos J. Garrido

Reviewer #2 (Remarks to the Author):

I review for the second time this interesting and well documented manuscript. I consider that you take into account my remarks but also the remarks of the second reviewer.

I have only two very small suggestions :

line 22 page 1 : add The serpentinites , probably formed

Done, thanks.

in the figure 8 : add the vertical scale, concerning the serpentinitized layer S1 and S2 they are probably not continuous, as in slow spreading ridges, serpentinites are patchy at the sea floor

We have reduced extension of the serpentinite layers accordingly.